# Adaptation in cone photoreceptors contributes to an unexpected insensitivity of primate On parasol retinal ganglion cells to spatial structure in natural images

Zhou Yu[†], Maxwell H Turner[†], Jacob Baudin, Fred Rieke*

Department of Physiology and Biophysics, University of Washington, Seattle, United States

**Abstract** Neural circuits are constructed from nonlinear building blocks, and not surprisingly overall circuit behavior is often strongly nonlinear. But neural circuits can also behave near linearly, and some circuits shift from linear to nonlinear behavior depending on stimulus conditions. Such control of nonlinear circuit behavior is fundamental to neural computation. Here, we study a surprising stimulus dependence of the responses of macaque On (but not Off) parasol retinal ganglion cells: these cells respond nonlinearly to spatial structure in some stimuli but near linearly to spatial structure in others, including natural inputs. We show that these differences in the linearity of the integration of spatial inputs can be explained by a shift in the balance of excitatory and inhibitory synaptic inputs that originates at least partially from adaptation in the cone photoreceptors. More generally, this highlights how subtle asymmetries in signaling – here in the cone signals – can qualitatively alter circuit computation.

*For correspondence:
rieke@uw.edu

[†]These authors contributed equally to this work

## Editor's evaluation

This study provides strong evidence that adaptation dynamics in cone photoreceptors of the primate retina can subtly change the balance of excitatory and inhibitory inputs to On parasol ganglion cells and thereby fundamentally affect how these cells integrate visual information. This provides important mechanistic insight into the previous observation that On parasol cells display nonlinear spatial stimulus integration under standard reversing gratings but linearly integrate signals in the context of natural scenes.

## Introduction

Components of neural circuits often transform neural signals nonlinearly. Common nonlinear relations include those between a sensory stimulus and the response of a primary sensory receptor (*Hudspeth and Corey, 1977*; *Baylor et al., 1987*), between presynaptic voltage and synaptic release (*Katz and Miledi, 1967*; *Huang and Neher, 1996*), and between membrane potential and action potential generation (*Hodgkin and Huxley, 1952*). Neural computation relies on the judicious control of these nonlinearities – in some cases to make overall circuit behavior near linear despite sharp deviations from linearity in the underlying components (*Werblin, 2010*). In some instances, linear circuit behavior emerges because signals are small and the underlying circuit mechanisms are not modulated sufficiently strongly to reveal their nonlinearities. In others, multiple nonlinear mechanisms act cooperatively to produce linear circuit behavior. Understanding how such control is exerted and when a neural

circuit operates linearly or nonlinearly has important consequences for constructing models of the nervous system and can provide a strong constraint for what kind of processing a circuit performs.

Retinal output neurons (i.e., retinal ganglion cells or RGCs) have long been classified by whether or not they linearly integrate signals across space (reviewed by *Field and Chichilnisky, 2007*; *Sanes and Masland, 2015*). Although this binary classification of spatial sensitivity is oversimplified, it has proven quite useful. The responses of spatially linear RGCs are proportional to the total light incident on their receptive field, whereas spatially nonlinear RGCs are also sensitive to the spatial distribution of light within the receptive field (*Enroth-Cugell and Robson, 1966*; *Hochstein and Shapley, 1976*). Classical tests of spatial integration used temporally modulated grating stimuli. These stimuli allow a clear prediction to be made for a linear RGC: a spatially linear RGC should produce no response because the light and dark bars that form the grating are equal and opposite in contrast and hence should cancel upon summation over space. RGCs that respond to gratings exhibit nonlinear spatial integration; such spatial nonlinearity can be explained by a nonlinearity at the output synapses of the bipolar cells presynaptic to a RGC that causes responses to the light and dark bars not to cancel (*Demb et al., 1999*; *Demb et al., 2001*; *Borghuis et al., 2013*; *Figure 1A*).

In a simple model like that in *Figure 1A*, the bipolar cell synaptic nonlinearities that shape responses to temporally modulated gratings would also predict sensitivity to spatial structure in other stimuli. However, many stimuli encountered by the retina have more complex spatial and temporal structures than periodic spatial gratings. Further, retinal circuits contain nonlinear circuit elements in addition to bipolar cell synapses. These two observations suggest that insights about spatial integration from grating stimuli may not generalize well to other stimuli. Indeed, we find that while periodic stimuli like contrast-reversing gratings elicit nonlinear spatial integration, nonperiodic stimuli like the onset of a spatial grating or a natural image often elicit near-linear spatial integration in On parasol RGCs (*Turner and Rieke, 2016*). Here, we explore the mechanistic basis and functional consequences of this stimulus dependence of spatial integration. We find that: (1) natural inputs and the onset of a spatial grating recruit large inhibitory synaptic input to On parasol RGCs that cancels spatial nonlinearities in excitatory synaptic input and suppresses sensitivity of the spike output to spatial structure. (2) Adaptation in the cone photoreceptors creates subtle asymmetries in the circuits controlling On parasol excitatory and inhibitory synaptic inputs. These asymmetries, magnified by circuit nonlinearities, can largely account for the differences in spatial integration among different stimulus types. And, (3) differences in spatial sensitivity of On and Off cells may enhance population coding of specific image features at the expense of others.

## Results

We begin with evidence that the insensitivity of responses of On parasol RGCs to spatial structure in natural inputs originates in the integration of excitatory and inhibitory synaptic inputs. We then show that adaptation in the cone photoreceptors plays a central role in controlling the balance of excitatory and inhibitory inputs, and that differences in On parasol responses to gratings and natural inputs can be largely explained by differences in the way that cones adapt to these stimuli. Finally, we explore how the difference in spatial integration between On and Off parasol RGCs impacts the encoding of specific stimulus properties.

### On parasol RGCs show stimulus-dependent spatial integration

A classic test for nonlinear spatial integration is to measure the response to a periodically modulated spatial grating restricted to the RGC receptive field center. Typically such measurements are made in 'steady state', well after grating onset. Nonlinear spatial integration causes a response at twice the temporal frequency of modulation – that is a 'frequency-doubled' or F2 response (*Enroth-Cugell and Robson, 1966*; *Hochstein and Shapley, 1976*). As observed previously, responses of both On and Off parasol RGCs showed strong frequency-doubled responses to contrast-reversing gratings (*Petrusca et al., 2007*; *Crook et al., 2008*; *Cafaro and Rieke, 2013*; *Figure 1B*, right gray box). Excitatory synaptic inputs and spike outputs exhibited similar F2 responses across a wide range of grating contrasts (*Figure 1B*, right, plots the F2 response normalized by the response to a modulated spatially uniform spot). The F2 responses are consistent with a nonlinearity in the bipolar output signals which creates spatial subunits in the On parasol receptive field (*Figure 1A*).

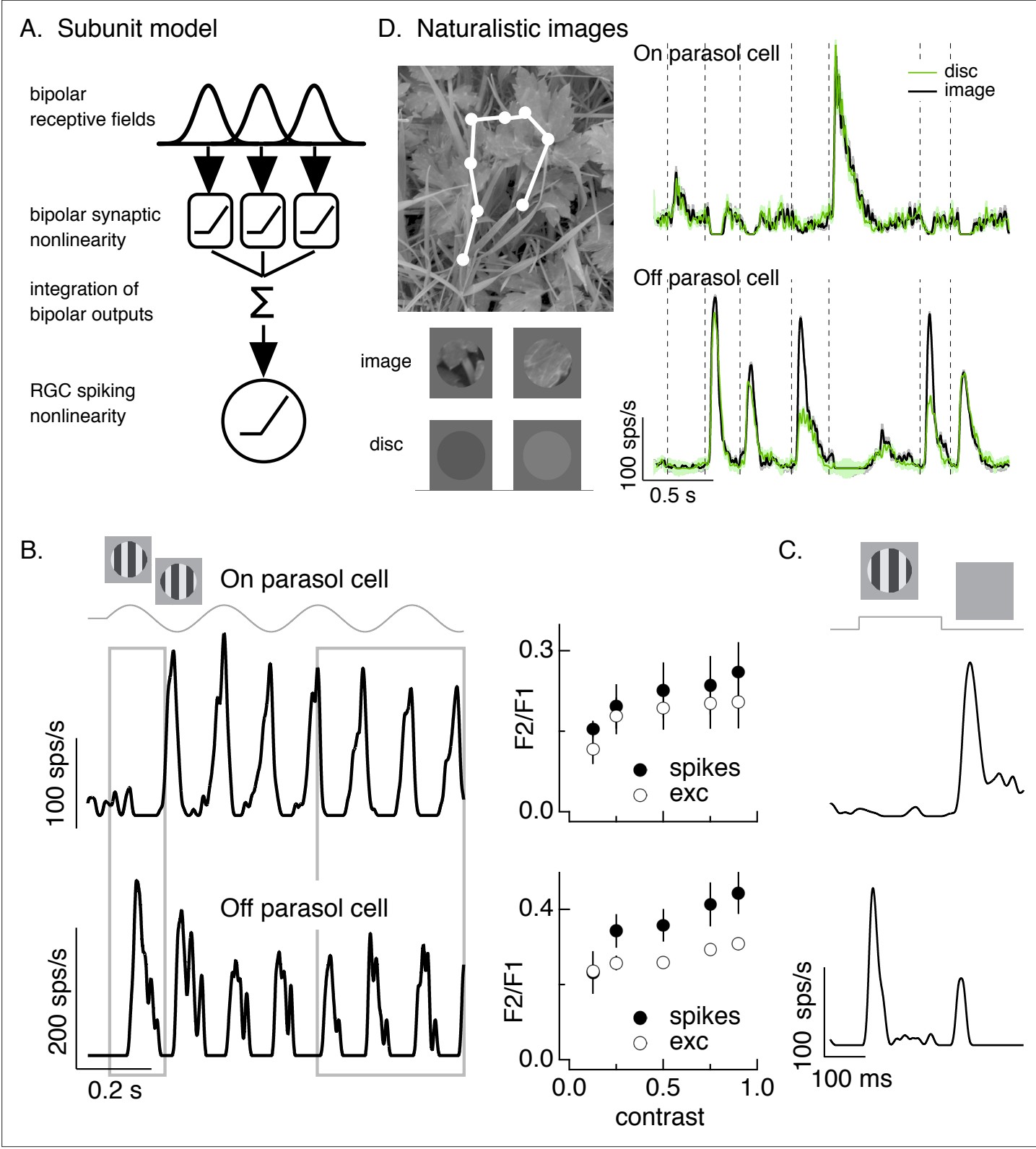

**Figure 1.** Differences in spatial integration for gratings and naturalistic stimuli. (**A**) Standard subunit model often used to account for nonlinear spatial integration by retinal ganglion cells (RGCs). (**B**) (Left) On and Off parasol RGC spike responses to contrast-reversing gratings. Both RGC types exhibit strong responses well after grating onset (right gray box), but responses at grating onset differ (left gray box). (Right) Dependence of frequency-doubled (F2) response on contrast for both excitatory inputs and spike output. The strength of nonlinear spatial integration was summarized as the ratio of

*Figure 1 continued on next page*

*Figure 1 continued*

the frequency doubled or F2 response measured for a split-field grating to the F1 response measured for a modulated spot. (**C**) On and Off parasol cell responses to flashed gratings differ. (**D**) On but not Off parasol cells show near-linear responses to natural movies. Top left shows image with eye movement trajectory in white. Bottom left shows two natural image movie frames and corresponding linear-equivalent discs. (Right) Black trace shows responses to a natural movie generated using the DOVES database (*Van Der Linde et al., 2009*). Movies were restricted to the receptive field center with a circular aperture. Green shows responses to a linear-equivalent movie, in which spatial structure within the receptive field center was replaced with a uniform disc with an intensity equal to the weighted average intensity within the receptive field center. The weighting was determined by a gaussian fit to the dependence of the response on the size of a test spot (see *Turner and Rieke, 2016* for details).

The online version of this article includes the following source data for figure 1:

**Source data 1.** Source data for *Figure 1B*.

The model in *Figure 1A*, however, failed to predict On parasol responses to spatial structure in several other stimuli. First, the onset of a spatial grating produced very different responses in On and Off parasol cells. This was true for a periodic grating (*Figure 1B*, left gray box) or a flashed grating (*Figure 1C*). Specifically, Off parasol cells, as predicted from the model in *Figure 1A*, responded at the onset of a periodic grating and at both the onset and offset of a flashed grating. On parasol cells, however, responded minimally at the onset of a periodic or flashed grating and strongly at the offset of a flashed grating. Thus, surprisingly, On parasol cells do not behave according to the model in *Figure 1A* at the onset of a grating stimulus, but do exhibit nonlinear spatial integration for subsequent phases of a modulated grating.

Second, On parasol cells responded weakly to spatial structure in natural inputs (*Figure 1D*). As in previous work (*Turner and Rieke, 2016*; *Turner et al., 2018*), we probed spatial integration of natural images by comparing responses to spatially structured natural inputs with responses to spatially uniform stimuli with temporal variations chosen to match those of the natural inputs (see below). We designed these stimuli to explore responses to naturalistic spatial and temporal structure in the receptive field center; there are clearly other aspects of real visual inputs that they do not capture.

Natural movie stimuli mimicked the retinal input during free viewing by translating a natural image across the retina according to measured human eye movements (*Van Der Linde et al., 2009*; *Figure 1D*, top left). The linearity of spatial integration was tested by comparing responses to natural movies with those to a linear-equivalent disc movie. The intensity of each frame of the linear-equivalent disc movie was calculated by linearly integrating the intensity of the corresponding frame of the natural movie weighted by an estimate of the cell's linear receptive field center. By construction, a cell that integrates inputs linearly over the receptive field center would produce identical responses to the original image and the disc. Both naturalistic and linear-equivalent stimuli were restricted to the receptive field center using an aperture with a size determined for each recorded cell from the dependence of response amplitude on spot size (*Turner and Rieke, 2016*; see Materials and methods). *Figure 1D* (bottom left) shows example images and corresponding linear-equivalent discs.

As reported previously, Off parasol RGCs responded differently to natural and linear-equivalent movies (green and black traces in *Figure 1D*, right). This difference indicates that Off parasol RGCs respond nonlinearly to spatial structure in natural visual inputs, as is expected from their responses to contrast-reversing and flashed gratings. On parasol RGCs, however, often showed similar responses to natural and linear-equivalent movies (*Figure 1D*, right; see *Turner and Rieke, 2016* for quantification); thus, unlike their responses to temporally modulated gratings, On parasol RGCs integrated spatial structure within their receptive field center linearly or near linearly for many natural inputs. Spatial integration in On parasol responses varied considerably across different naturalistic stimuli, even within the same cell (*Turner and Rieke, 2016*, *Freedland and Rieke, 2021*). This further suggests that the spatial and/or temporal structure present in a natural movie can shape the spatial integration properties of On parasol cells.

The results summarized in *Figure 1* demonstrate that spatial integration in On parasol RGCs exists on a stimulus-dependent spectrum from linear integration (e.g., for the onset of grating stimuli or for most natural movies) to nonlinear integration (e.g., for subsequent phases of a modulated grating). This spectrum is not explained by differences in image contrast: temporally modulated gratings elicit F2 responses across a broad range of contrasts (*Figure 1B*, right), including those encountered in natural scenes (see Materials and methods), and even high contrast gratings fail to elicit a response at onset. To understand how On parasol cells can behave according to the model in *Figure 1A* under

some stimulus conditions but not others, we first identify the synaptic input properties that control spatial integration in these cells.

## On parasol excitatory inputs elicited by natural inputs are more nonlinear than spike outputs

Interpreting responses to natural image movies like those in *Figure 1D* is complicated by possible history dependence – for example, the response at a given time could reflect the onset of a new image feature and/or the offset of a previous feature. To simplify the correspondence between image features and responses, we turned to flashed image patches (*Figure 2*). All patches were flashed starting from the same gray background, and hence differences in response can be attributed to the structure in the patch itself rather than the past history of the stimulus. To focus on patches that elicited spike responses from On parasol RGCs, we selected patches with a higher luminance in the receptive field center compared to the gray background.

For each cell, we recorded spike responses, excitatory synaptic inputs, and inhibitory synaptic inputs elicited by a set of image patches and corresponding linear-equivalent discs. *Figure 2A–C* shows responses of one example cell for 30 patches from a single image. Traces at the top show the time course of responses to two example patches, and the main panels plot mean (± standard error of the mean, SEM) for responses to six repeats of each individual patch. Responses were quantified by integrating the response over the time that the stimulus was presented. Consistent with the responses to time-varying naturalistic stimuli in *Figure 1D*, On parasol RGCs generated similar spike responses to flashed natural images and linear-equivalent discs (*Figure 2A*); this indicates linear or near-linear spatial integration.

Two aspects of the responses to flashed image patches suggested that inhibitory synaptic input contributes to the linearity of On parasol RGC spike responses. First, two image patches that elicited similar amplitude excitatory input could elicit quite different spike responses; for example, the black traces at the top of *Figure 2A, B* are from the same two image patches. The excitatory inputs for these patches were similar in amplitude (*Figure 2B*, top), but the example patch on the left elicited a smaller spike response than the patch on the right (*Figure 2A*, top). This is consistent with inhibitory input suppressing the response to the patch on the left relative to that on the right. Second, excitatory inputs elicited by many image patches exceeded responses to linear-equivalent discs, for example the black and green responses at the top of *Figure 2B* and the cluster of points below the unity line in the bottom panel. The difference in image and disc responses indicates nonlinear spatial integration. Spike responses of the same cell to the same image patches are much more linear, as indicated by the similarity of the black and green traces at the top of *Figure 2A* and the similarity of the image and disc responses across patches illustrated in the bottom panel.

*Figure 2A,B* shows that spatial integration of On parasol excitatory inputs differs from that of spike output – at least for this cell and image. To test the generality of this difference across cells and images, we summarized the response to each patch with a nonlinearity index (NLI). This index measures the normalized difference between responses to the image and disc

$$NLI = (R_{image} - R_{disc})/(R_{image} + R_{disc}),  \quad (1)$$

where $R_{image}$ is the response to the image and $R_{disc}$ the response to the disc. We averaged NLIs across all patches probed for a given cell and image – resulting in a single value for excitatory synaptic input and one for spike output for each cell. The resulting average NLIs varied considerably across different cells and images (*Figure 2D*; each point corresponds to a different cell), and some images systematically produced positive NLIs for both spike output and excitatory input (see also *Turner and Rieke, 2016*). Across cells, however, the NLI for excitatory inputs was significantly larger than that for spike output (*Figure 2D*; 0.25 ± 0.05 vs 0.08 ± 0.07, mean ± SEM, p < 0.01, *n* = 10). This is not due to large NLIs in a small subset of image patches; instead, more than half of the patches from a typical image showed nonlinear spatial integration for excitatory inputs (e.g., points below the unity line in the patch-by-patch analysis of *Figure 2B*) and near-linear spatial integration for spike outputs (e.g., points on or near the unity line in *Figure 2A*).

The difference in the sensitivity of On parasol excitatory input and spike output to spatial structure in natural images differs markedly from the situation for periodic gratings, in which both excitatory

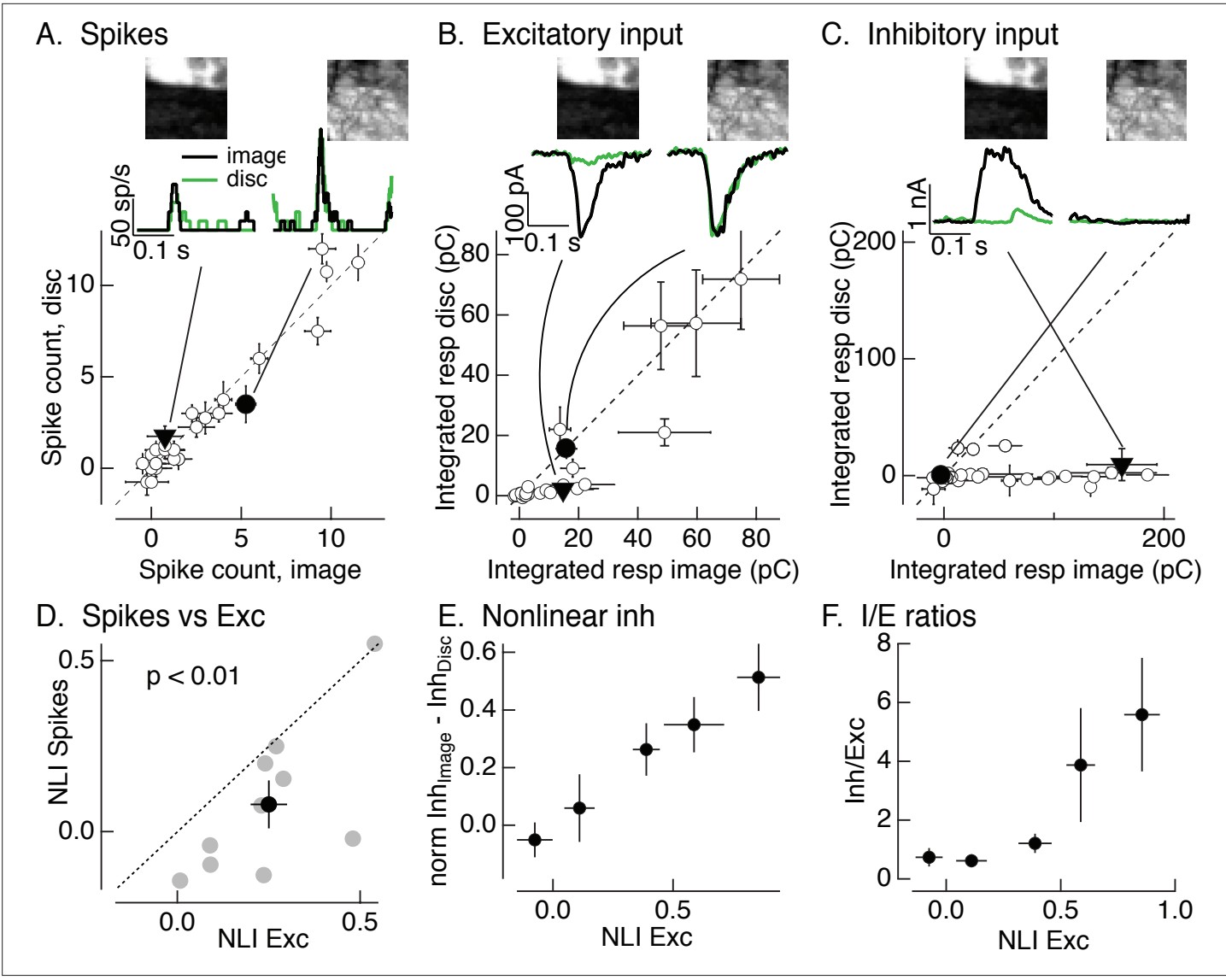

**Figure 2.** Synaptic integration leads to unexpected linearity of On parasol responses to natural images. (**A**) Spike count for responses of one On parasol retinal ganglion cell (RGC) to image patches and corresponding linear-equivalent stimuli. Each stimulus was flashed for 250 ms and spikes counted during that time. Peristimulus time histograms for the two highlighted points are at the top along with the corresponding images; the black trace is the response to the image patch and the green trace is the response to the linear-equivalent stimulus. (**B**) Excitatory synaptic inputs for the same cell and image patches as in A. Responses are the integrated current during the stimulus presentation. The same two image patches are highlighted. (**C**) Inhibitory synaptic input for the same cell and image patches. (**D**) Excitatory inputs are more nonlinear than spike output. Comparison of nonlinearity index (NLI; see *Equation 1*) for spike response with that for excitatory input for 10 On parasol cells (gray) and mean across cells (black, mean ± standard error of the mean [SEM]). For each cell, NLIs were averaged across all image patches probed. (**E**) Patches that recruit nonlinear excitatory input also recruit nonlinear inhibitory input. Nonlinear inhibitory input (image response − disc response) plotted against excitatory NLI. Excitatory NLIs were separated into five bins, and the nonlinear inhibitory input for the image patches in each bin was averaged across patches and cells (*n* = 8). For each cell, inhibitory input was normalized by the largest input produced across patches. (**F**) Ratio of inhibitory input to excitatory input increases for patches that elicit strong nonlinear excitatory input. Excitatory NLIs were binned as in *E*, and the average *I*/*E* ratio (for 8 cells) was computed for all patches in each bin.

The online version of this article includes the following source data for figure 2:

**Source data 1.** Source data for *Figure 2A–F*.

inputs and spike output exhibit clear nonlinear spatial integration (*Figure 1B*, right). We explore the origin of this difference below.

## Natural stimuli recruit strong inhibitory input to ganglion cells

A logical hypothesis that follows from the results above is that natural images recruit strong inhibitory synaptic input to On parasol cells that cancels nonlinear excitatory input. The results described below support this hypothesis. We will then return to the issue of why periodic grating stimuli fail to recruit similar inhibitory synaptic input.

*Figure 2C* shows inhibitory inputs in response to the same image patches in the same example cell as the spikes and excitatory inputs in *Figure 2A, B*. Plotted points show the mean (± SEM) integrated response from six repeats of each patch. Inhibitory synaptic input exhibited strongly nonlinear spatial integration, as indicated by the points that fall below the unity line in *Figure 2C*. Image patches that elicit similar excitatory input can elicit very different inhibitory input; for example, the filled points and corresponding black traces in the top panels of *Figure 2B, C*. Further, this difference corresponds to the difference in spike response (black traces in *Figure 2A*). Patches that elicit highly nonlinear excitatory input can also elicit highly nonlinear inhibitory input (e.g., the filled triangles in *Figure 2B, C*) – that is, spatial structure in natural image patches recruits both excitatory and inhibitory input. These observations are consistent with cancelation of nonlinear excitatory input by inhibitory input and the insensitivity of the spike response to spatial structure in patches that elicit nonlinear excitatory input (e.g., the filled triangle and corresponding responses in *Figure 2A*).

*Figure 2A–C* shows results from one example cell. To test the generality of these observations, we quantified the relationship between excitatory and inhibitory input across patches from multiple images and RGCs. To effectively cancel nonlinear excitatory input, inhibitory input should overlap in time with excitatory input and have at least two additional properties. First, nonlinear inhibitory input should be recruited by the same image patches that elicit nonlinear excitatory input (as in the example patches at the top of *Figure 2B, C*). To test this, we compared the NLI for excitatory input with the difference between the (normalized) inhibitory input elicited by the image and disc (we did not use NLIs for inhibitory input because they tended to be either 0 or 1, obscuring any graded dependence on excitatory NLI). We divided the NLIs for excitatory input for individual patches into five discrete bins, and then measured the average difference in inhibitory input elicited by the image and linear-equivalent disc across all RGCs and patches for each excitatory NLI bin. *Figure 2E* plots the results; each point represents the mean (± SEM) difference in inhibitory input plotted against the mean (± SEM) excitatory NLI. Nonlinear inhibitory input increased systematically with increasing excitatory NLI. Thus, patches that elicited nonlinear excitatory input also elicited nonlinear inhibitory input.

Second, inhibitory input should be sufficiently large to effectively cancel nonlinearities in excitatory input. To test this, we measured the ratio of the amplitude of inhibitory input to excitatory input (the *I/E* ratio) as a function of the nonlinearity in excitatory input (*Figure 2F*). We again averaged inhibitory/excitatory ratios across image patches and RGCs. *Figure 2F* plots this average *I/E* ratio (mean ± SEM) against the excitatory NLI, organized into the same discrete bins as *Figure 2E*. The electrical driving force on excitatory synaptic input is about threefold larger than that on inhibitory synaptic input near the threshold for spike generation. Because of this difference in driving force, inhibitory inputs (measured as currents at the reversal potential for excitatory inputs) need to be at least threefold larger than excitatory inputs to have a substantial impact on spike output. Patches that elicited near-linear excitatory input corresponded to *I/E* ratios <1; in this case, inhibitory input is unlikely to substantially affect spike output. Patches that elicited strongly nonlinear excitatory input, however, corresponded to *I/E* ratios >3; for these patches inhibitory input is sufficiently large to impact spike output.

Collectively, the results of *Figure 2* indicate that (1) excitatory synaptic input to On parasol cells exhibits larger spatial nonlinearities than spike outputs, and (2) inhibitory synaptic input has the required properties to compensate for nonlinearities in excitatory input and account for the linearity of the spike response. We next characterize inhibitory input in more detail and investigate why it does not linearize responses to grating stimuli as it does responses to natural images.

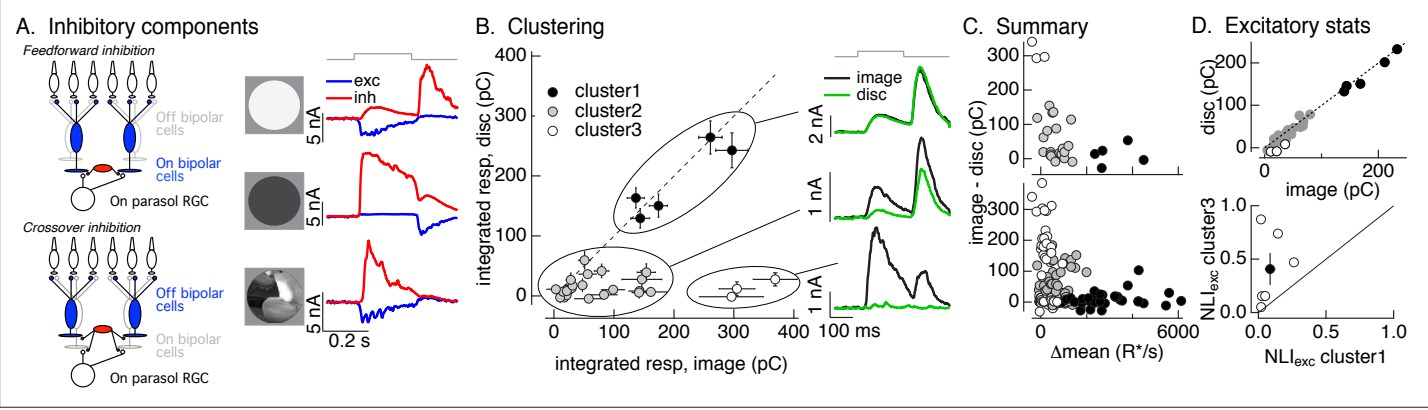

**Figure 3.** Identifying the components of inhibitory input to On parasol cells elicited by natural image patches. (**A**) (Left) Circuits responsible for two components of inhibitory synaptic input to On parasol retinal ganglion cells (RGCs). (Right) The onset of a light increment (top) produces an increase in excitatory input (blue) and a small, delayed increase in feedforward inhibitory input (red). The onset of a light decrement (middle) produces a decrease in excitatory input and a large increase in crossover inhibitory input. Different regions of spatially structured inputs (bottom) can produce large crossover inhibitory input that coincide with increases in excitatory input. (**B**) Clustering of inhibitory synaptic inputs elicited by image patches. Clustering was based on response time course. Panels at right show average responses to the image (black) and corresponding linear-equivalent disc (green) for each cluster. Cluster 1 is dominated by feedforward inhibitory input and shows linear spatial integration (i.e., image and disc responses are near identical). Cluster 3 is dominated by crossover inhibitory input and shows strongly nonlinear spatial integration. Cluster 2 is intermediate between the two others. (**C**) Summary of relation between nonlinear inhibitory input (image–disc) and mean luminance for each of the clusters from B (top), and summary across cells (bottom). (**D**) (top) Excitatory responses to image patches and corresponding linear-equivalent discs for each cluster from B. (bottom) Comparison of nonlinearity indices (see *Equation 1*) for excitatory inputs corresponding to clusters 1 and 3 for six cells (open circles) and mean across cells (closed circle).

The online version of this article includes the following source data and figure supplement(s) for figure 3:

**Source data 1.** Source data for *Figure 3B–D*.

**Figure supplement 1.** PCA-based clustering applied to contrast–response data.

## Crossover inhibitory input linearizes on parasol responses

Inhibitory synaptic input to On parasol RGCs has two distinct components that originate from different retinal circuits (*Figure 3A*, left; *Cafaro and Rieke, 2013*; *Crook et al., 2014*). Feedforward inhibitory input originates from activity in On retinal pathways and, like excitatory input, is elicited by increases in light intensity (*Figure 3A*, top left). Crossover inhibitory input originates in Off retinal pathways and is elicited by decreases in light intensity (*Figure 3A*, bottom left). This means that a spatially uniform increase in light intensity increases both excitatory input and feedforward inhibitory input (*Figure 3A*, top right), while a spatially uniform decrease in intensity decreases excitatory input and increases crossover inhibitory input (*Figure 3A*, middle right).

Inhibitory synaptic input contributes minimally to On parasol spike responses to spatially uniform stimuli (*Cafaro and Rieke, 2013*). The increase in feedforward inhibitory input elicited by a spatially uniform increment is not sufficiently large to have a sizable impact on spike responses (*I/E* ratios are generally less than 1). And the decrease in excitatory input alone is sufficient to eliminate spiking for a spatially uniform decrement, and hence crossover inhibitory input makes little contribution. However, different regions of spatially structured stimuli can elicit different types of synaptic input – for example bright regions of a stimulus can cause an increase in excitatory input while dark regions can cause an increase in crossover inhibitory input (*Figure 3A*, bottom right). This can cause the (potentially large) crossover inhibitory input to overlap in time with excitatory input and hence impact spike output (*Cafaro and Rieke, 2013*).

Different natural image patches elicited inhibitory input that varied widely in amplitude and time course. The linearity of spatial integration similarly varied widely across patches; in some cases, responses to the image patch and the linear-equivalent disc were similar, while in others they were very different. These observations suggest that the contributions of feedforward and crossover inhibitory input differed across different patches. Testing this suggestion required estimating the contributions of each component of inhibitory input to the response to a given image patch. To do this, we used a principal components-based approach to cluster inhibitory responses elicited by different

image patches (see Materials and methods); clustering used only the time course of the responses to the flashed image patches, and did not use any information about the patch itself or responses to the linear-equivalent discs. We set the target number of clusters to three as this was the minimum needed to separate responses with clearly different time courses. The goal of this analysis was to organize the responses according to their time course, and the discrete clusters were a matter of convenience only and do not reflect discreteness in the actual responses.

The three clusters defined by this approach had consistently different spatial integration properties. *Figure 3B* (left) compares inhibitory synaptic inputs elicited by flashed image patches and corresponding linear-equivalent discs for each response in each cluster for an example RGC. *Figure 3B* (right) shows the time course of average image and disc responses in each cluster. Responses in the first cluster had a time course consistent with feedforward inhibitory input (see *Figure 3A*, top right) and showed linear spatial integration – that is responses to image patches and linear-equivalent discs were near identical (*Figure 3B*, top right). Responses in the third cluster had a time course consistent with crossover inhibitory input and exhibited strongly nonlinear spatial integration – with much larger responses to images than to the corresponding linear-equivalent discs (*Figure 3B*, bottom right). Responses in the second cluster were smaller in amplitude and had more variable time courses and spatial integration properties than the others.

As a control, we applied the clustering procedure used for images to responses to increment and decrement spots. We retained the cluster definitions determined for the images – that is responses to increment and decrement spots were assigned to the clusters that were defined from the time course of responses to flashed image patches. High contrast increment spots, which elicit feedforward inhibitory input, led to responses in the first cluster, intermediate contrast increments led to responses in the second cluster, and decrement spots, which elicit crossover inhibitory input, led to responses in the third cluster (*Figure 3—figure supplement 1*). Hence, the clustering approach separates responses associated with known increases and decreases in light intensity: cluster 1 is dominated by feedforward inhibitory input, cluster 3 is dominated by by crossover inhibitory input, and cluster 2 is a mixture of the two.

We next compared the properties of the image patches corresponding to each inhibitory cluster. We started with how the difference between the inhibitory response to the image and disc – that is, the magnitude of spatially nonlinear inhibitory input – depended on the mean intensity of the image patch. As expected from the stimulus dependence of feedforward and crossover inhibitory input (e.g., *Figure 3A*), image patches eliciting responses in the first cluster were bright compared to the image mean (closed circles in *Figure 3C*); flashing these patches results in a net increase in light intensity in the receptive field center. Image patches in the third cluster had a mean intensity similar to the image mean (open circles in *Figure 3C*), and hence flashing these patches resulted in little change in light intensity in the receptive field center. Image patches in cluster 2 were again intermediate (gray circles in *Figure 3C*). The distinctions among clusters held across cells and images (*Figure 3C*, bottom; each point represents measurements of responses to a single image patch from a single RGC). We did not probe image patches with large negative intensity because we focused on patches that elicited an increase in spike rate (as in *Figure 2*). What is clear from this analysis, however, is that patches with little change in mean intensity – that is those corresponding to cluster 3 – can elicit large and spatially nonlinear crossover inhibitory input.

We next identified the excitatory synaptic inputs corresponding to the first and third inhibitory clusters – that is, those that we could associate with feedforward and crossover inhibitory input. *Figure 3D* (top) plots the excitatory inputs elicited by linear-equivalent discs against those elicited by the corresponding image patch for the same cell and patches as *Figure 3B, C*; symbols correspond to the inhibitory clusters defined in *Figure 3B*. The excitatory inputs differed systematically for different inhibitory clusters. Excitatory responses to patches corresponding to the first cluster were large and exhibited minimal spatial nonlinearity (*Figure 3D*, closed circles). Excitatory responses corresponding to the third cluster were relatively small and exhibited strong spatial nonlinearities (*Figure 3D*, open circles). Excitatory responses corresponding to the second cluster had intermediate properties (*Figure 3D*, gray circles).

To extend this comparison to multiple RGCs and images, we compared the mean NLI for excitatory inputs corresponding to inhibitory clusters 1 and 3. This collapses the excitatory inputs for each cell into two numbers: the mean NLI for patches corresponding to inhibitory cluster 1 and the

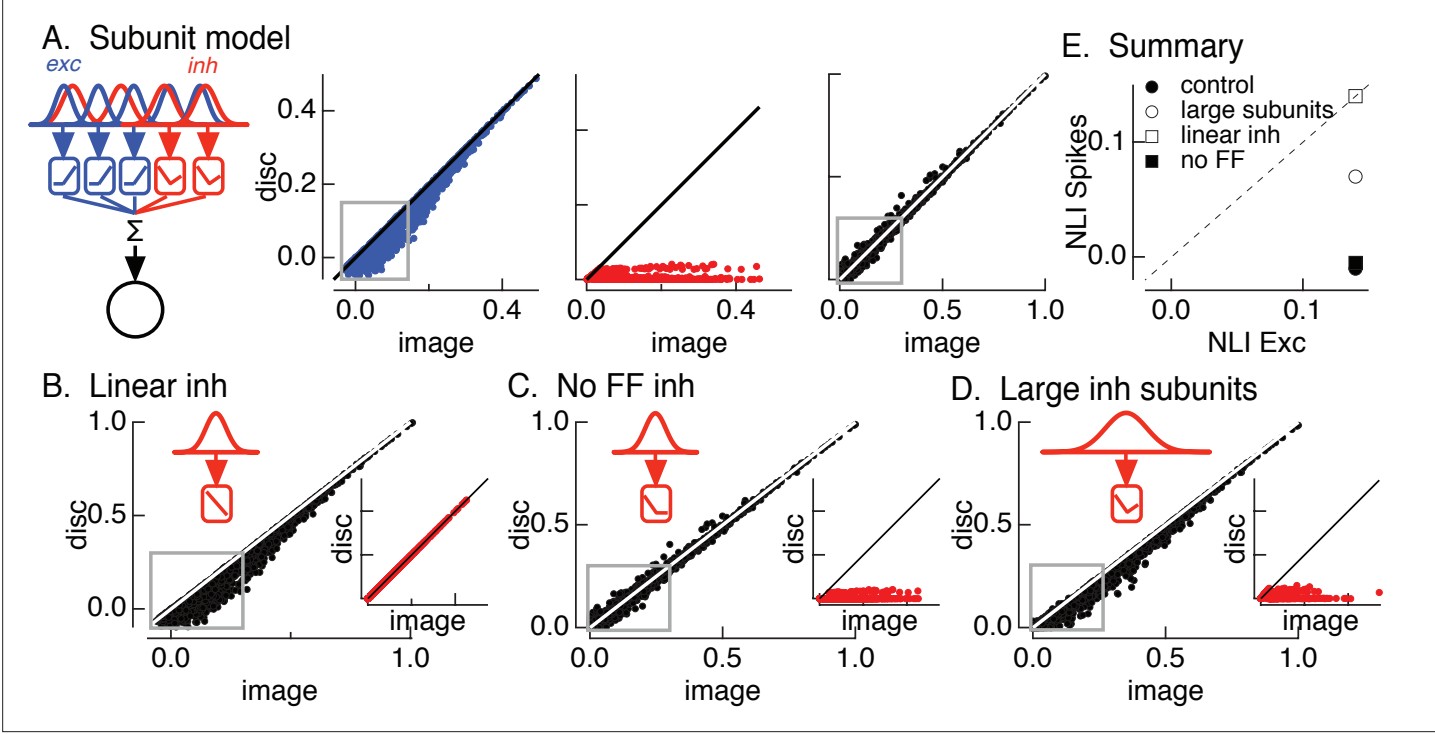

**Figure 4.** Crossover inhibitory synaptic input is necessary and sufficient for linear spatial integration. (**A**) Model construction. (left) Architecture of subunit models. Parallel pathways generated excitatory (blue) and inhibitory (red) synaptic input to a retinal ganglion cell (RGC). Each pathway was composed of multiple subunits, each with a separate spatial filter and output nonlinearity. Excitatory and inhibitory subunits covered the same region of space but were positioned independently. (right) Excitatory inputs (blue), inhibitory inputs (red), and spike outputs (black) for 'default' model parameters (see Materials and methods). Gray boxes highlight patches that exhibit the strongest nonlinear spatial integration in excitatory inputs. (**B**) Spike output for model in which inhibitory synaptic input integrates linearly over space. (**C**) Spike output for model lacking feedforward inhibitory synaptic input. Feedforward inhibitory input was eliminated by changing the shape of the nonlinearity in the inhibitory pathway (inset). (**D**) Spike output for model in which the spatial filters for inhibitory subunits were doubled in size and the density of inhibitory subunits was correspondingly decreased. (**E**) Summary of difference in nonlinearity index (NLI) for spikes and excitatory input for several models for inhibitory input (see also *Figure 2D*).

The online version of this article includes the following figure supplement(s) for figure 4:

**Figure supplement 1.** Contrast–response functions and fits.

mean NLI for patches corresponding to inhibitory cluster 3. We then plotted these against each other (*Figure 3D*, bottom; open circles are individual cells, closed circle is mean ± SEM). Across cells, the NLIs for excitatory responses corresponding to cluster 3 were larger than those for patches corresponding to cluster 1.

In summary, different natural image patches elicited inhibitory responses with different contributions of feedforward and crossover inhibitory input, and correspondingly different spatial integration properties. In particular, crossover inhibitory input showed strongly nonlinear responses to spatial structure, and was elicited by image patches that also elicited strongly nonlinear excitatory inputs. These observations suggest that crossover inhibitory input cancels nonlinear excitatory input and linearizes responses to natural inputs.

## Functional properties of inhibitory input required to cancel nonlinear excitatory input

What functional properties of inhibitory input are required for effective cancellation of nonlinear excitatory input? It is difficult to selectively manipulate inhibitory input experimentally to answer this question due to off-target effects of pharmacological agents. Instead, we used subunit receptive field models with parallel excitatory and inhibitory paths (*Figure 4A*; see Materials and methods); these are an extension of the bipolar cell subunit receptive field models we used previously to account for Off parasol responses to natural images (*Turner and Rieke, 2016*). These models allowed us to

investigate how sensitivity to spatial structure in the input depended on specific properties of the inhibitory circuitry.

The first stage of the model filters spatial input through two regularly spaced grids of subunits – with separate subunits for excitatory and inhibitory paths. The receptive field size of each subunit was set to be consistent with responses to contrast-reversing gratings (*Turner and Rieke, 2016*). The filtered signal in each subunit was passed through a nonlinearity determined by the contrast–response function for either excitatory or inhibitory synaptic input (*Figure 4—figure supplement 1*). Inhibitory subunits had 'U'-shaped nonlinearities, reflecting both feedforward and crossover components. Excitatory (blue) and inhibitory (red) subunit outputs were weighted by a Gaussian profile representing the receptive field center and then summed, with a threefold larger weighting of excitatory inputs reflecting the larger driving force near spike threshold. Spike responses (black) were predicted by thresholding this summed signal to eliminate negative responses. These models focus exclusively on spatial integration and do not consider time.

*Figure 4A* shows modeled excitatory, inhibitory, and spike responses for a collection of natural image patches from a single image. This 'base' model captured the measured cancelation of nonlinear excitatory input by inhibitory input and the linearity of the spike response (*Figure 4A*, right). In particular, the model captured nonlinear spatial integration in excitatory inputs (indicated by the points falling below the unity line in *Figure 4A*, left – see gray box) and the linear spatial integration in the spike output for these same image patches (*Figure 4A*, right). We summarized spatial integration for a given image by averaging the NLI (*Equation 1*) across patches for both excitatory input and spike output. We repeated this analysis for each image and averaged the resulting NLIs. Consistent with experiment (*Figure 2D*), the NLI for excitatory input was larger than that for spike output (*Figure 4E*, closed circle). We did not attempt to make quantitative predictions of responses to specific patches because of uncertainties in model details such as the subunit positions in a recorded cell.

We next probed how three manipulations of inhibitory input affected model responses. First, we replaced the measured inhibitory subunit nonlinearity with a linear contrast–response function. This forces inhibitory input to integrate linearly across space. Inhibitory input in this case was unable to cancel nonlinear excitatory input (see gray box in *Figure 4B*), and the NLIs for excitatory input and spike output were similar (*Figure 4E*, open square). This result highlights the necessity of nonlinear spatial integration if inhibitory input is to cancel nonlinearities in excitatory input. Second, we eliminated feedforward inhibitory input by setting the positive contrast region of the contrast–response function for inhibitory input to zero (*Figure 4C*). This had minimal effect on the model spike response (compare *Figure 4A*, right and *Figure 4C*) and on the difference in the NLIs for excitatory input and spike output (*Figure 4E*, closed square). Hence, crossover inhibitory input alone was sufficient to cancel nonlinear excitatory input, and feedforward inhibitory input contributed little to shaping model responses (see also *Cafaro and Rieke, 2013*). Third, we doubled the size of the inhibitory subunits without changing the size of the excitatory subunits (*Figure 4D*). This substantially decreased the ability of inhibitory input to cancel nonlinear excitatory input (*Figure 4D*, gray square and *Figure 4E*, closed and open circles). This analysis suggests that linearity of the spike response requires that nonlinear subunits for excitatory and inhibitory circuits have similar sizes.

In summary, the modeling of *Figure 4* indicates that the insensitivity of the On parasol spike response to spatial structure in natural inputs depends on nonlinear crossover inhibitory input that has a similar spatial scale as nonlinear excitatory input and exhibits strong nonlinear spatial integration. Importantly for the analyses below, this means that Off retinal circuits are the main source of the inhibitory input that regulates spatial integration.

## Spatial integration of flashed vs periodic gratings differs

The experiments described thus far (1) indicate that the linearity of On parasol responses to spatial structure in natural images originates from recruitment of inhibitory input, and (2) identify key properties of inhibitory input that make it effective in linearizing spike responses. This description, however, does not explain why On parasol cells respond to spatial structure in some stimuli but not others (see *Figure 1*). The experiments and analyses described in the next two sections indicate that the periodic time course of classic grating stimuli substantially impacts synaptic integration by rapidly modulating the adaptational state of the cone photoreceptors. This time-dependent adaptation contributes

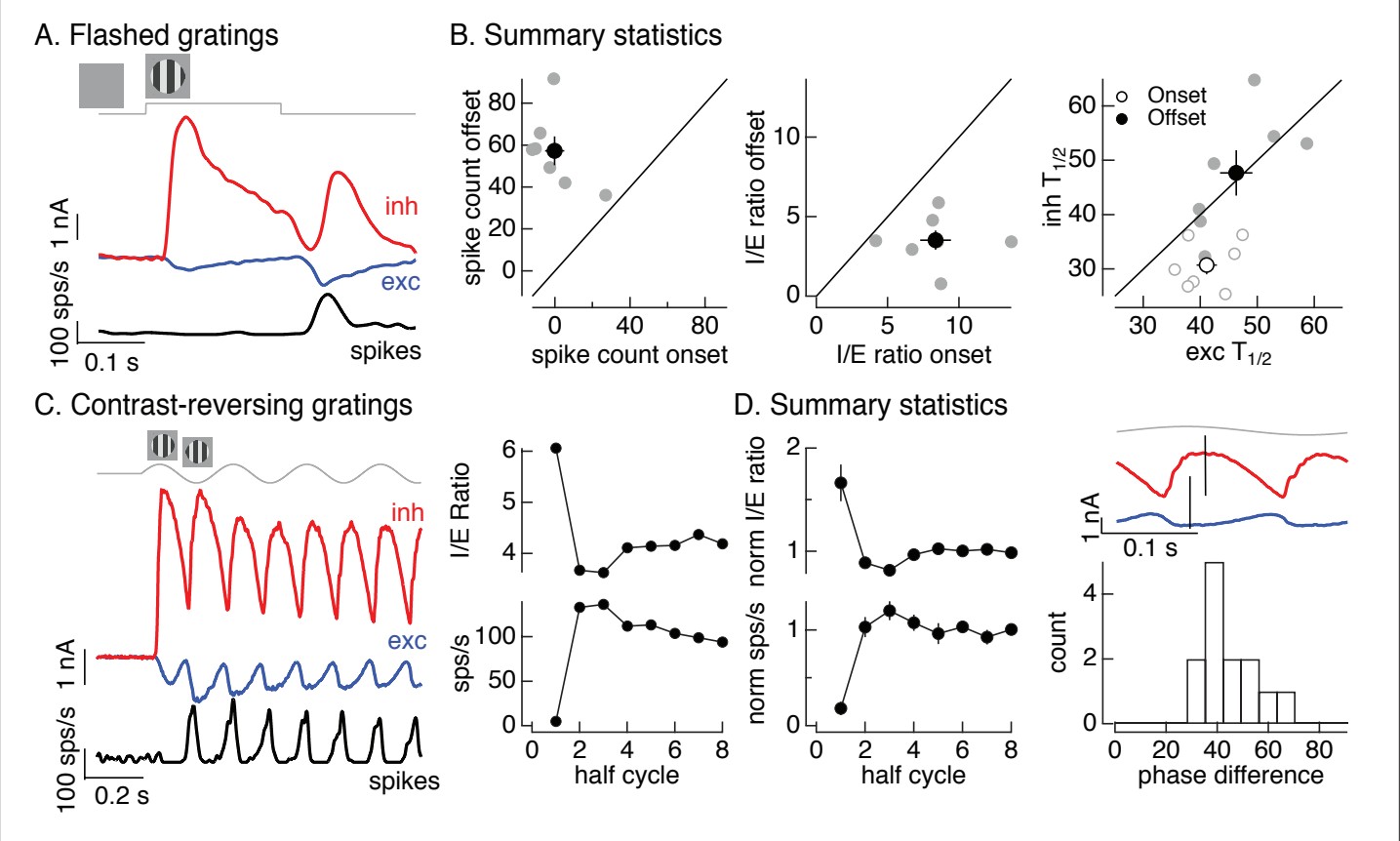

**Figure 5.** Time dependence of On parasol responses to flashed and contrast-reversing gratings. (**A**) Excitatory and inhibitory synaptic inputs and spike response to a flashed grating. (**B**) Summary of responses to flashed gratings from seven On parasol retinal ganglion cells (RGCs). (left) Spike count at grating offset vs that at grating onset. Individual cells are in gray, mean (± standard error of the mean, SEM) is in black. (middle) Ratio of inhibitory input to excitatory input at grating offset vs that at grating onset. (right) Relative timing of excitatory and inhibitory synaptic inputs, measured as the time to reach half-maximal amplitude. (**C**) (left) Excitatory and inhibitory synaptic inputs and spike output in response to a contrast-reversing grating modulated at 4 Hz. (right) *I/E* ratio and peak spike rate for each grating half-cycle. (**D**) Summary of statistics of grating responses across 13 On parasol RGCs. (left) *I/E* ratio (mean ± SEM) and spike rate (mean ± SEM) have been normalized by the mean value for a half-cycle number of 6 or more. (right) Histogram of time of peak excitatory input relative to inhibitory input. Response timing estimated by fitting each response with a sinusoid. For 4 Hz stimuli, a phase difference of 40° corresponds to 14 ms.

The online version of this article includes the following source data and figure supplement(s) for figure 5:

**Source data 1.** Source data for **Figure 5B–D**.

**Figure supplement 1.** Time dependence of Off parasol responses to contrast-reversing gratings.

strongly to responses to contrast-reversing gratings, such as those in **Figure 1B** and many other studies.

 **Figure 1C** introduced On parasol responses to a flashed grating. Here, we analyze these responses in more detail since they can be compared directly to the flashed images in **Figure 2**. The onset of a flashed grating elicited a minimal spike response, while the grating offset elicited a clear response (**Figure 5A**, black). To collect results across cells, we compared the number of spikes elicited at grating onset with that at grating offset (**Figure 5B**, left). The offset response exceeded the onset response in each recorded cell (gray points; onset response 0 ± 5 spikes, offset response 57 ± 7 spikes, mean ± SEM, $n = 7$, $p < 0.001$). To explore the origin of this difference, we compared excitatory and inhibitory inputs produced at grating onset and offset (**Figure 5A**). Both the relative magnitude and the timing of excitatory and inhibitory inputs differed for grating onset and offset. To quantify the difference in amplitude, we computed the *I/E* ratio based on the integrated inhibitory and excitatory currents at grating onset and offset. The *I/E* ratio was about twofold larger at grating onset than offset (**Figure 5B**, middle; $p < 0.01$). To quantify the difference in timing, we compared the time at which the excitatory

and inhibitory currents reached a half-maximal value ($T_{1/2}$). Excitatory input was delayed relative to inhibitory input at grating onset but not offset (*Figure 5B*, right; onset p < 0.01, offset p = 0.9). Both of these factors should contribute to the larger spike response at grating offset.

Like the difference in responses at the onset and offset of a flashed grating, responses to contrast-reversing gratings depended on time since grating onset (see *Figure 1B*). *Figure 5C, D* quantifies this dependence by integrating responses (spikes, inhibitory input, and excitatory input) during each half-cycle of the grating. The spike response during the first half-cycle of a contrast-reversing grating was substantially smaller than that during subsequent half-cycles (*Figure 5C*, left and bottom right). Similarly, the ratio of inhibitory input to excitatory input was substantially larger during the first half-cycle of the grating than during subsequent half-cycles (*Figure 5C*, left and top right). These features held across cells (*Figure 5D*, left, plots mean ± SEM, *n* = 13). Off parasol cells exhibited the opposite behavior, with the smallest *I/E* ratios and largest spike responses during the first half-cycle of a grating (*Figure 5—figure supplement 1*). This asymmetry between On and Off parasol responses is particularly interesting since Off circuits dominate inhibitory input to On parasol RGCs (*Figure 3*). We will pursue this On/Off difference in the next section.

The kinetics of excitatory and inhibitory synaptic inputs in response to a modulated grating also differed. Excitatory and inhibitory input increased with similar time courses during the first half-cycle of the grating, but for subsequent half-cycles increases in excitatory input led increases in inhibitory input (*Figure 5D*, top right, shows excitatory and inhibitory synaptic currents from the same cell as *Figure 5C*). The firing rate was highest in the time window in which excitatory inputs had increased but inhibitory input was still small (e.g., compare black and blue traces in *Figure 5C*, left). We quantified this time shift across cells by measuring the phase difference between sinusoidal fits to excitatory and inhibitory currents (*Figure 5D*, bottom right). The measured phase differences correspond to time shifts of 10–15 ms.

The analyses summarized in *Figure 5* highlight the dependence of nonlinear spatial integration on the stimulus time course. Nonlinear spatial integration is weak or absent at the onset of a grating because inhibitory input is large compared to excitatory input and the two overlap in time. Nonlinear spatial integration increases for later cycles of a periodic grating stimulus, due to a decrease in the *I/E* ratio and a delay of inhibitory input relative to excitatory input.

## Cone adaptation shapes synaptic integration

Nonlinear spatial integration in excitatory synaptic input is typically generated by rectifying nonlinearities at the bipolar output synapse (*Demb et al., 1999*; *Demb et al., 2001*; *Borghuis et al., 2013*; but see *Schreyer and Gollisch, 2021*). The results summarized in *Figures 2–4* indicate that the impact of this spatial nonlinearity on spike output can be regulated by inhibitory synaptic input. The results in the previous section identify stimulus dynamics as a key regulator of the excitatory/inhibitory balance and hence the strength of nonlinear spatial integration. This section explores the origin of this dependence on stimulus dynamics.

The time course of the light intensity encountered by a single receptive-field subunit differs substantially between the onset and offset of a flashed grating and likewise between the first and subsequent half-cycles of a modulated grating. Prior to the onset of either stimulus type, subunits encounter the mean stimulus intensity, while at later time points they undergo larger dark–light or light–dark transitions. Several time-dependent nonlinear mechanisms could cause this difference in stimulus time course to contribute to the responses illustrated in *Figure 5*. One such mechanism is adaptation in the cone photoreceptors, which we explore below.

Measured cone photocurrents to sinusoidal stimuli showed three features that could potentially shape RGC responses (*Figure 6A*): (1) the amplitude of the cone response was smaller on the first half-cycle of the grating than on subsequent cycles; (2) cone responses were asymmetric, with larger responses to the dark phase of the grating (decrements) than to bright phase (increments); and, (3) the kinetics of cone responses to dark-to-light transitions were sped relative to responses to light-to-dark transitions. These properties are apparent in both the cone current (*Figure 6A*, top) and voltage (*Figure 6A*, bottom) responses. These properties of the cone responses are created by rapid adaptation in the phototransduction process (*Angueyra et al., 2022*), due to a light-dependent increase in PDE activity (*Nikonov et al., 2000*), which causes the gain of the cone light response to be modulated during each cycle of the sinusoid. The cone responses are well captured by an empirically derived

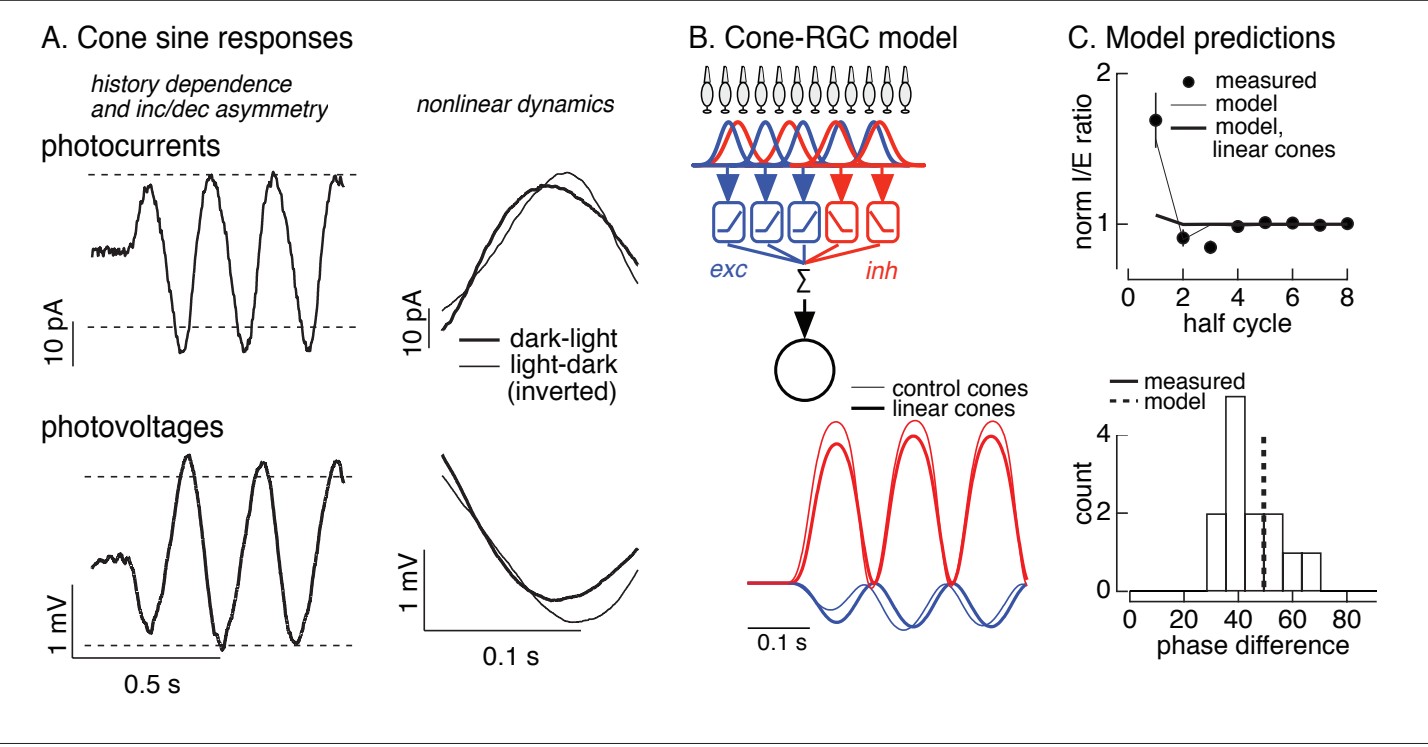

**Figure 6.** Nonlinearities in cone responses to contrast-reversing gratings and implications for On parasol synaptic inputs. (**A**) Cone phototransduction currents elicited by contrast-reversing gratings exhibit three properties that could shape the time course of retinal ganglion cell (RGC) responses: (1) a dependence on stimulus history, (2) an asymmetry between light increments and decrements, and (3) a difference in kinetics of cones responding to light-to-dark vs dark-to-light transitions. Photocurrents are average responses to 5 Hz, 75% contrast sinusoidal stimuli from six cones. Photovoltages are average responses to 4 Hz, 50% contrast sinusoidal stimuli from 16 cones. Dashed horizontal lines are displaced equally above and below the mean response to highlight the asymmetry between increment and decrement responses. (**B**) Circuit model to explore how properties of cone responses from A could alter RGC responses. The model from *Figure 4* was adapted to incorporate a first stage based on models for the cone responses (*Figure 6—figure supplement 1*). Subunit nonlinearities were set by measured contrast–response functions. Shown are examples of excitatory and inhibitory synaptic inputs predicted by models with adapting and linear cone models. (**C**) Comparison of model predictions with experiment. (top) Histogram of I/E ratio from On parasol recordings (from *Figure 5D*) and predictions from models with adapting (thin line) and nonadapting (thick line) cones. (bottom) Histogram of phase difference between excitatory and inhibitory synaptic inputs elicited by contrast-reversing gratings (from *Figure 5D*) and prediction from model (dashed vertical line).

The online version of this article includes the following figure supplement(s) for figure 6:

**Figure supplement 1.** Cone model and responses to sinusoidal stimuli.

**Figure supplement 2.** Measured and predicted excitatory synaptic inputs in responses to contrast-reversing gratings across a range of frequencies.

biophysical model of cone phototransduction (see *Figure 6—figure supplement 1*, Materials and methods and *Angueyra et al., 2022* for details). This model allows us to predict and manipulate cone responses to contrast-reversing gratings.

Different aspects of light stimuli control excitatory and inhibitory synaptic inputs to On parasol cells: light increments dominate excitatory inputs, whereas light decrements dominate inhibitory inputs (*Figure 3A*). This suggests that the asymmetry in the amplitude and timing of increment and decrement responses in the cones could control the relative magnitude and timing of excitatory and inhibitory synaptic inputs to an On parasol RGC and hence how these inputs are integrated to control spike output. The difference in the responses of On and Off parasol cells to the onset of a grating (*Figures 1B, C and 5C* and *Figure 5—figure supplement 1*) supports this suggestion.

To test for a role of cone adaptation in controlling spatial integration, we modified the subunit model from *Figure 4* to take cone signals as input (see Materials and methods). Hence, we simulated responses of an array of cones to contrast-reversing gratings, and then used these simulated cone responses as input to models with excitatory and inhibitory subunits, including measured contrast–response functions. We used this approach to predict responses to gratings for models incorporating

both adapting and nonadapting (i.e., linear) cones (*Figure 6B*; *Figure 6—figure supplement 2*; see Materials and methods for model details). The linear cone model was implemented by filtering the stimulus with a linear filter fit to the cone response to a brief, low-contrast flash. Note that the measured RGC contrast–response functions should capture time-independent (i.e., static) nonlinearities in the cone responses, particularly the asymmetry between the amplitude of increment and decrement responses; these contrast–response functions, however, will not capture time-dependent nonlinearities in the cone responses. Hence, this modeling specifically predicts how the time-dependent nonlinearities introduced by adaptation in the cones impact RGC responses.

Models with adapting cones predicted a 70–80% larger *I/E* ratio on the first grating cycle compared to the steady-state ratio (*Figure 6C*, top, thin line). This time dependence of the *I/E* ratio was very similar to that observed in RGC synaptic inputs (closed circles) and was absent in predictions of models with linear cones (thick line). Models with adapting cones also captured the temporal delay between excitatory and inhibitory input (thin lines in *Figure 6B*; dashed line in *Figure 6C*, bottom); this delay was again absent for models with linear cones (thick lines in *Figure 6B*). Excitatory input is dominated by cones undergoing a dark-to-light transition, whereas inhibitory input is dominated by cones undergoing a light-to-dark transition. The timing difference in the model originates from the temporal asymmetry in the responses of cones encountering increases and decreases in light level (*Figure 6A*, right). This difference in cone kinetics is a property of adaptation and is absent in linear models of the cone responses and in the RGC models with linear cone inputs. The light-to-dark vs dark-to-light difference in the kinetics of the cone responses is absent in the first half-cycle of a periodic grating, because all cones begin from the same mean intensity, but present in subsequent cycles when different cones start from different adaptational states.

The modeling summarized in *Figure 6* supports the following picture for the time dependence of responses to periodic grating stimuli. Prior to grating onset, all cones are in a uniform adaptational state. The onset of the grating produces an increase in both excitatory and inhibitory synaptic input to an On parasol cell, and these occur with near-identical kinetics (*Figures 5A, C and 6B*). Rapid cone adaptation (time constant <100 ms) limits the amplitude of the increase in excitatory input at grating onset (compare thick and thin lines in *Figure 6B*). The uniform adaptational state of the cones, however, is altered by the grating. Hence, for subsequent cycles of the grating, cones encountering an increase in light intensity start from a higher signaling gain and respond more quickly than cones encountering a decrease in light intensity. This increases the amplitude of excitatory input relative to inhibitory input, and creates a time window in which excitatory input increases before inhibitory input. Without adaptation (i.e., with linear cones), this shift in amplitude and timing of excitatory input relative to inhibitory input does not occur.

## Removing nonlinearities in cone responses reduces nonlinear spatial integration

To directly test the hypothesis that cone adaptation shapes On parasol RGC responses to contrast-reversing gratings, we used the cone model to design grating-like stimuli that minimized the impact of adaptation on cone responses. In the case of a sinusoidal stimulus like a contrast-reversing grating, this means identifying a stimulus time course for which the cone model predicts a sinusoidal response. To achieve this, we transformed a grating stimulus to minimize the difference between the response of the adapting cone model to the transformed stimulus and the response of a linear cone model to the original stimulus (*Figure 7A*). We refer to this procedure as the 'light-adaptation clamp' (see Materials and methods). Since a linear cone responds sinusoidally to a sinusoidal input, the transformed stimulus we seek is one for which the adapting cone model predicts a sinusoidal response. Cone responses to the modified grating stimuli are predicted to exhibit less initial history dependence and symmetrical responses to increments and decrements (in both dynamics and amplitude).

We tested the stimuli constructed as illustrated in *Figure 7A* in recordings from voltage-clamped cones (*Figure 7B*, top). Across recorded cones, the transformed stimuli altered cone responses as predicted by the model (*Figure 7B*). First, the kinetics of the response to the dark-to-light transition was slowed (closed arrowhead in *Figure 7B*, top). Second, the difference in amplitude of the response to the first vs second cycle was reduced (*Figure 7B*, bottom left). Third, the increment/decrement asymmetry was reduced (open arrowhead in *Figure 7B*, top; *Figure 7B*, bottom right). Hence the

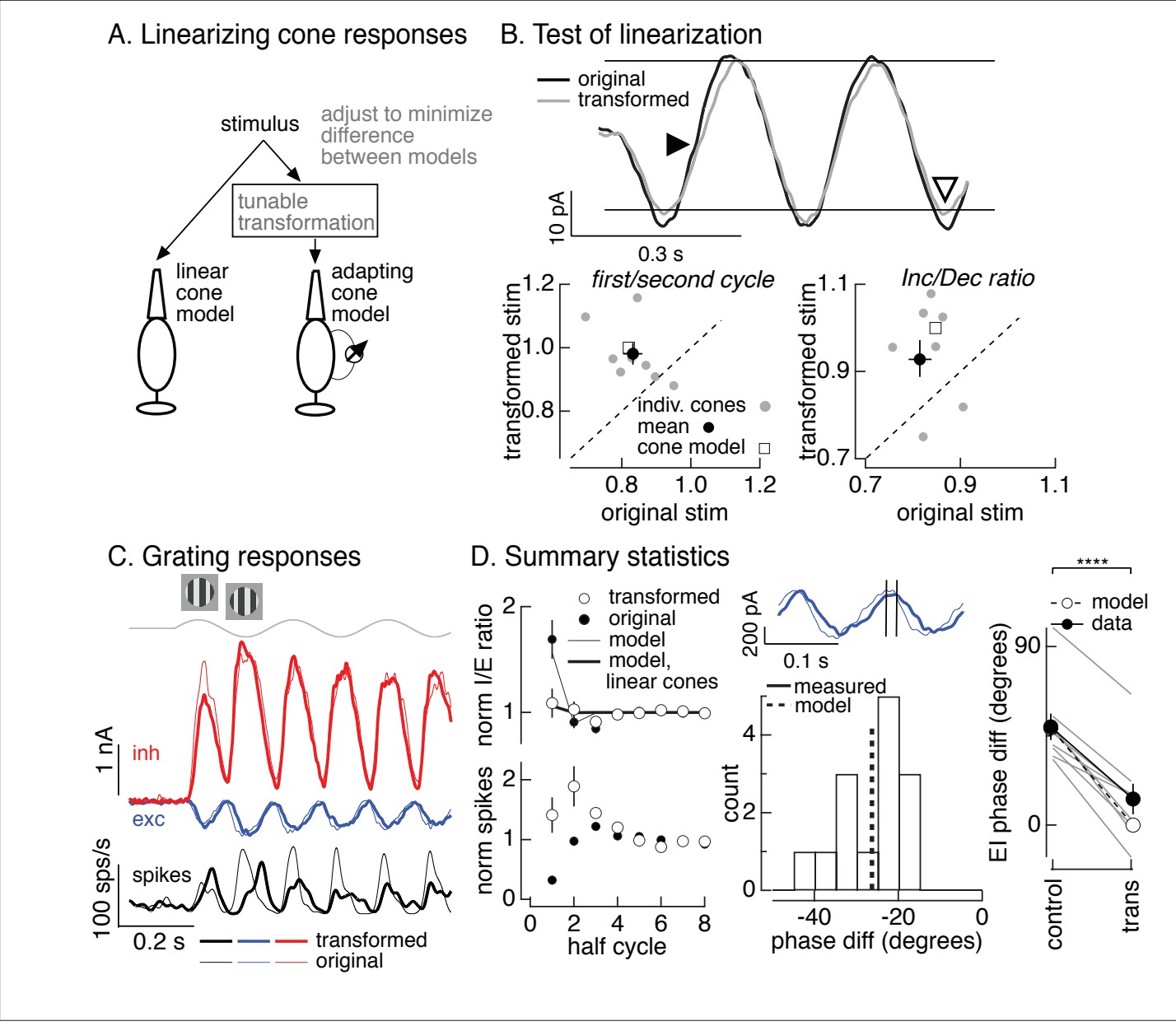

**Figure 7.** Minimizing cone adaptation minimizes time dependence of On parasol responses to contrast-reversing gratings. (**A**) Approach to minimize nonlinearities in cone responses. Standard grating stimuli were transformed to minimize the difference between the modeled responses of an adapting cone responding to the transformed grating and a linear (nonadapting) cone responding to the original grating. (**B**) Test of procedure from A. (top) Measured cone responses to original (black) and transformed (gray) grating stimuli. (bottom) Summary across recorded cones. Responses to the transformed stimulus showed less history dependence (left; ratio of the amplitude of the measured response on the first cycle of the grating to that on the second cycle) and less of an increment/decrement asymmetry (right). Open squares show predictions from adapting cone model. (**C**) On parasol responses to original (thin traces) and transformed (thick traces) grating stimuli. (**D**) Summary of responses to original and transformed gratings from nine On parasol cells. (left) I/E ratios and spike counts have been normalized in each cell by the responses for grating half-cycles >6. (middle) Histogram of difference in timing of excitatory input in response to original and transformed gratings. A phase shift of 30° corresponds to a 21-ms difference in timing. (right) Difference in timing of excitatory and inhibitory synaptic inputs for original and transformed gratings and prediction from model of *Figure 6B* (open circles and dashed line, **\*\*\*\*** denotes p < 0.0001 for difference in phase).

The online version of this article includes the following source data for figure 7:

**Source data 1.** Source data for *Figure 7B–D*.

light-adaptation clamp procedure successfully identified stimuli that minimized the impact of adaptation on cone responses.

The light-adaptation clamp procedure provided a tool to test the impact of adaptation in cone phototransduction on responses of On parasol RGCs to periodic gratings. Specifically, if cone adaptation contributes to the small spike response of On parasol RGCs during the onset of a contrast-reversing grating (*Figure 5C, D*), then On parasol responses to the modified stimuli should show less time dependence. This was indeed the case (*Figure 7C*, thick traces are responses to the transformed stimuli and thin traces are responses to the original grating). First, the decrease in *I/E* ratio observed for the original grating was largely absent for the transformed grating (*Figure 7D*, top left). The time dependence of the *I/E* ratios for original and transformed stimuli agreed well with predictions from the models of *Figure 6B* with linear or nonlinear cone inputs (*Figure 7D*, top left). Second, the suppression of the spike response at the onset of the original grating stimulus was absent for the transformed stimulus (*Figure 7D*, bottom left).

As illustrated in *Figure 6A*, adaptation also causes cones to respond faster to dark-to-light vs light-to-dark transitions. Correspondingly, minimizing cone adaptation changed the relative timing of excitatory and inhibitory inputs to On parasol cells (*Figure 7D*, middle and right). Excitatory inputs in response to the transformed stimuli were substantially delayed compared to excitatory inputs in response to the original stimuli (*Figure 7D*, middle). This shift in timing of excitatory input caused it to overlap more with inhibitory input, and the substantial phase shift between excitatory and inhibitory input observed for conventional gratings was almost entirely eliminated by the transformed gratings (*Figure 7D*, right). The difference in timing of excitatory and inhibitory inputs for both grating types was well predicted by the models in *Figure 6B* based on linear or nonlinear cones (dashed line and open circles in *Figure 7D*, right). These results indicate that the temporal differences in cones exposed to increments and decrements (*Figure 6A*) plays a central role in determining the kinetics of excitatory and inhibitory input and hence the effectiveness with which inhibitory input cancels excitatory input.

The combination of modeling in *Figure 6* and direct manipulation of the cone responses in *Figure 7* indicate that cone adaptation can explain the majority of the observed time dependence of the On parasol responses to contrast-reversing gratings. The decrease in *I/E* ratio after grating onset and the speeding of excitatory input relative to inhibitory input account for the strength of the steady-state responses to contrast-reversing gratings, and both of these depend on adaptation in the cones. Downstream time-dependent nonlinearities (e.g., depression at the bipolar output synapse; *Ke et al., 2014*) could also contribute, but such contributions appear small compared to cone adaptation (see Discussion). These results add an unexpected element to classic subunit models such as those in *Figure 1A*: the impact of nonlinearities at the bipolar output synapse on RGC spike responses is controlled by how the RGC integrates excitatory and inhibitory input, and this *E/I* balance is in turn shaped by adaptation in the cone photoreceptors.

## Cone adaptation and circuit nonlinearities can explain linear responses to natural images

Natural inputs vary over time due to eye movements and object motion within an image. Why then do On parasol RGCs respond weakly to spatial structure in time-varying natural inputs (*Figure 1D*) despite their strong responses to periodic grating stimuli? To answer this question, we used the cone/subunit model described above to predict how cone adaptation impacts the balance of excitatory and inhibitory inputs for On parasol cell responses to time-varying natural inputs.

Stimuli were created by sampling a static natural image with a simulated eye movement trajectory. These stimuli are identical to the movies used in *Figure 1* except for the use of simulated rather than real eye movements. Eye movements were simulated by a diffusional process to account for fixational eye movements, interrupted by occasional large and discrete changes in position to account for saccades (see Materials and methods for details). The resulting spatiotemporal pattern of inputs was converted to signals in the modeled cone photoreceptor array (*Figure 8A*, bottom, shows modeled photocurrent responses of a single cone). This was repeated for adapting and linear cone models (adapting cone models followed *Figure 6—figure supplement 1*; the linear cone model consisted of a linear filter convolved with the stimulus). Adaptation substantially attenuated the amplitude of modeled cone responses compared to responses of cones without adaptation (*Figure 8A*, bottom).

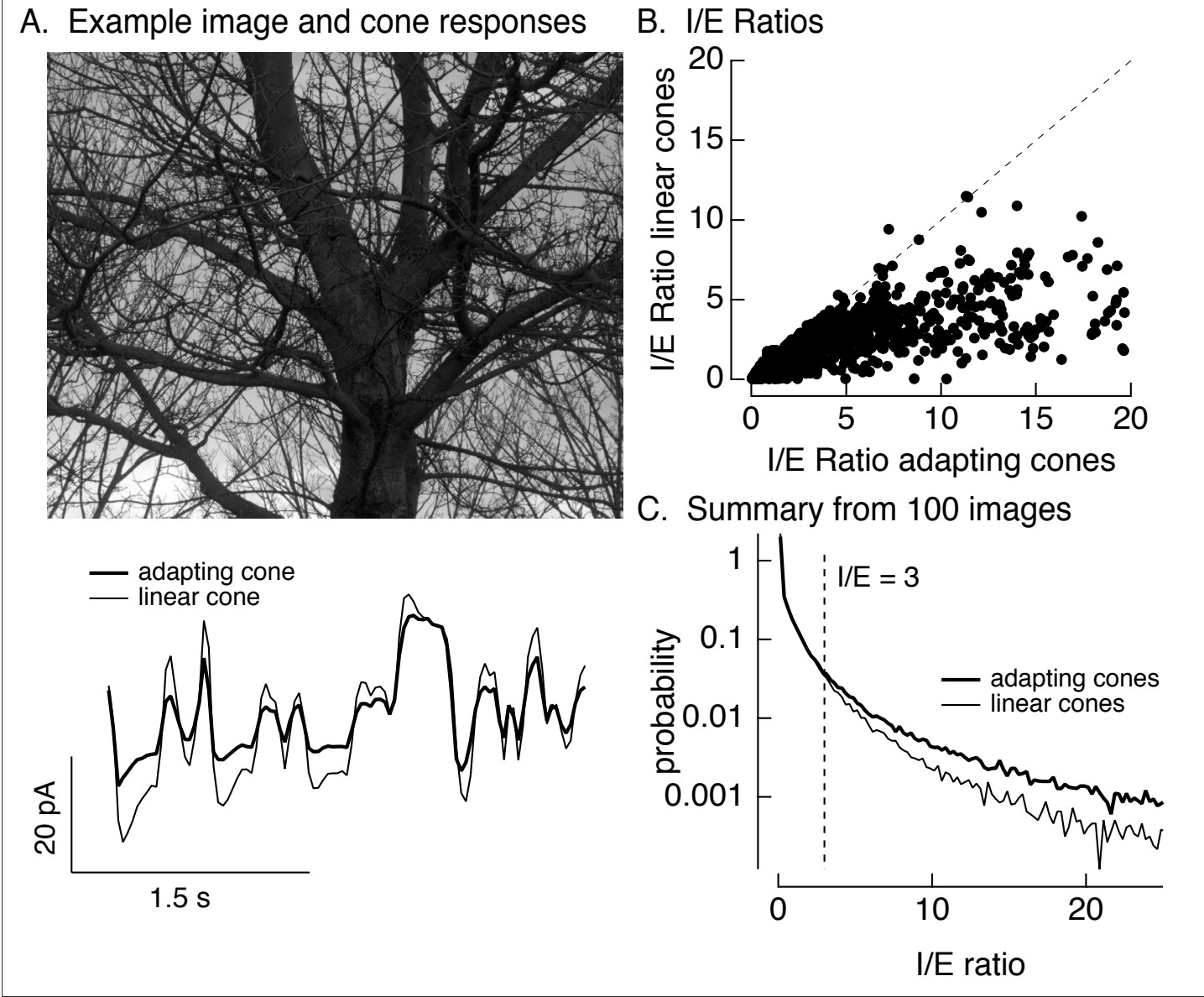

**Figure 8.** Cone adaptation substantially increases *I/E* ratio for natural images. (**A**) The model from *Figure 6B* with either adapting or nonadapting cones was used to predict excitatory and inhibitory synaptic inputs during eye movements about a scene. Predicted cone photocurrents from the adapting and linear cone models are shown below the image. (**B**) *I/E* ratios were computed for every 30 ms time window for models incorporating adapting and linear cones. Ratios for linear cones were systematically smaller than those for adapting cones (i.e., most points fall below the unity line). (**C**) Histogram of *I/E* ratios, calculated as in B, for many images.

We used the modeled adapting and nonadapting cone signals as input to subunit models for excitatory and inhibitory synaptic inputs as in *Figure 6B*. We used these models to predict *I/E* ratios in every 30-ms time bin (*Figure 8B*). These predicted *I/E* ratios are sensitive to changes both in the relative timing and in the amplitude of excitatory and inhibitory input. *I/E* ratios varied widely across time bins, but were systematically larger for models with adapting cones compared to models with nonadapting cones (*Figure 8B*). To test for a similar effect across images and trajectories, we constructed histograms of *I/E* ratios for models with adapting and nonadapting cones; *Figure 8C* compares histograms from many images, each with an independent eye movement trajectory. Cone adaptation substantially increased the number of patches in which inhibitory input was sufficiently large to attenuate excitatory input (*I/E* ratios exceeding 3).

This analysis suggests that cone adaptation shapes responses to contrast-reversing gratings and time-varying natural inputs quite differently. This difference can be explained by the different luminance trajectories encountered by cones during these stimuli, and the corresponding differences in the time course of cone adaptation. In the case of a sinusoidally modulated grating, an increase in intensity at a given spatial location always occurs after a decrease, and likewise decreases occur after increases. Hence, a cone encountering an increase in intensity starts from a state in which signaling gain is high (due to the preceding decrease in intensity and relief of adaptation), and a cone encountering a increase in intensity starts from a low gain state (due to the preceding decrease in intensity and engagement of adaptation). This both speeds and increases the amplitude of responses of cones signaling intensity increases relative to those signaling decreases (see *Figure 6A*) – which in the case of an On parasol cell means shifting signaling in favor of excitatory synaptic inputs over inhibitory synaptic inputs (see *Figure 6B*).

Naturalistic inputs typically do not show such predictable behavior: intensity increases are not systematically preceded by low intensity regions of a scene. Instead, on average, cones encountering decreasing of increasing intensity start from the average image intensity – much like the first half-cycle of a modulated grating or the onset of a flashed grating. In all such cases, cone adaptation had a strong impact on the *I/E* ratio and spike response (e.g., linear vs adapting cones in *Figures 6 and 8* and control vs transformed stimuli in *Figure 7*). Specifically, adaptation in the cones shifted the *I/E* balance in favor of inhibitory input, and this in turn decreased the sensitivity of On parasol RGC responses to spatial structure.

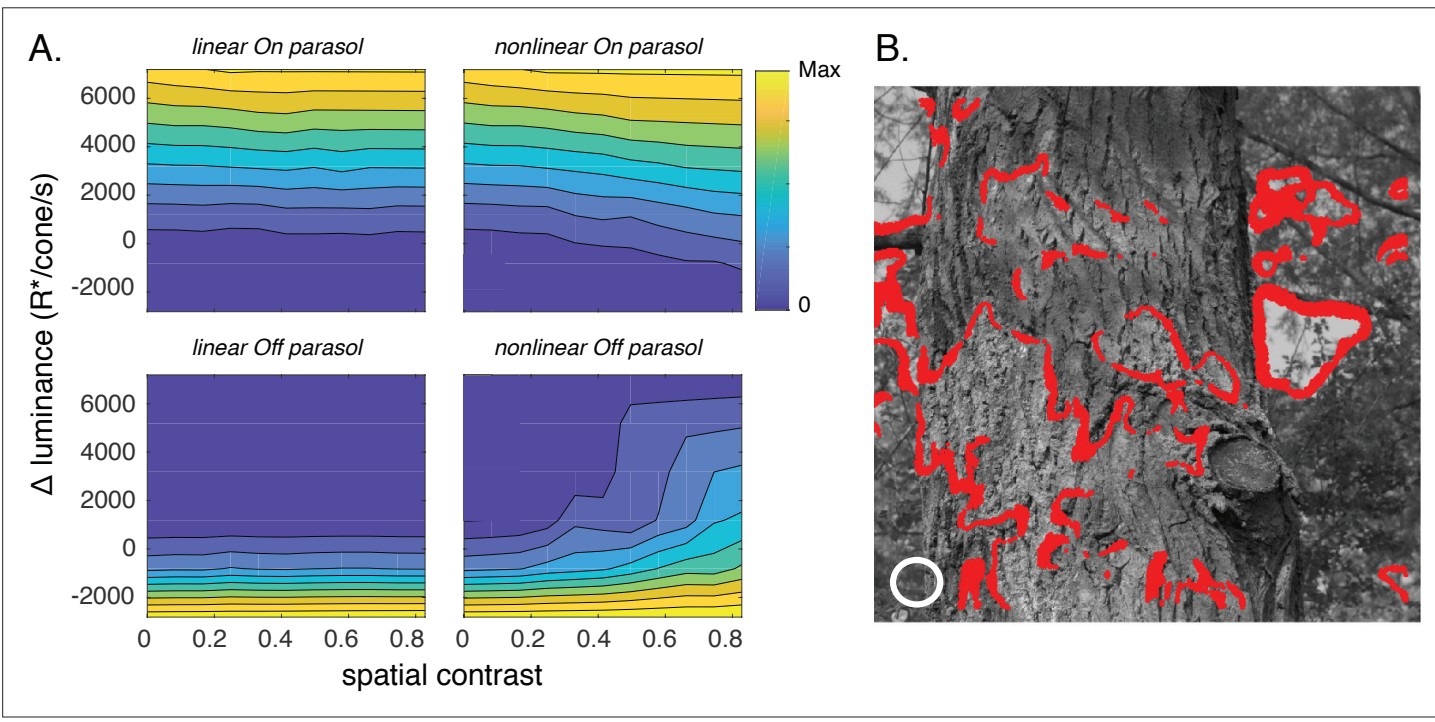

**Figure 9.** Joint activity of On and Off parasol retinal ganglion cells (RGCs) encodes patches with positive luminance and high spatial structure. (**A**) Responses of On and Off parasol models as a function of the spatial contrast (*x*-axis) and change in mean luminance (*y*-axis) for a collection of image patches. Panels show predicted constant–response contours for both On and Off parasol RGC models for linear and nonlinear spatial integration. Mean luminance and spatial contrast were computed for a simulated receptive field center with a size shown by the white circle in B. Spatial contrast was defined as the standard deviation divided by the mean of the pixel values in the receptive field center. (**B**) Image patches eliciting activity of both On and Off parasol cells for the linear On parasol and nonlinear Off parasol RGC models. These are in the upper right corner of the contour plots in A.

The online version of this article includes the following figure supplement(s) for figure 9:

**Figure supplement 1.** Discrimination of image patches with similar mean luminance (5000 R*/cone/s) but different spatial contrast for joint responses of On and Off parasol cells.

**Figure supplement 2.** On and Off parasol cell responses to the same collection of image patches for two On/Off pairs.

## Unique natural stimulus features coded by coactivation of On and Off cells

On and Off ganglion cells of the same type are classically assumed to respond to opposite polarity but otherwise identical stimulus features. Asymmetries between On and Off cells challenge this simple picture (*Chichilnisky and Kalmar, 2002*; *Zaghloul et al., 2003*; *Sagdullaev and McCall, 2005*; *Murphy and Rieke, 2006*; *Nichols et al., 2013*; *Turner and Rieke, 2016*; *Ravi et al., 2018*). The experiments and analysis below suggest that asymmetries in spatial integration (as in *Figure 1*) cause joint activation of On and Off parasol cells to encode image features not encoded by either cell type alone. This analysis and the associated conclusions are restricted to activation of the receptive field center.

We used subunit models to identify image patches predicted to coactivate On and Off parasol cells. We first sampled patches of natural inputs through a grid of subunits as in *Figure 4*. We then constructed two RGC models (see Materials and methods and *Turner and Rieke, 2016*): (1) in a spatially linear model, subunit outputs were weighted by a Gaussian receptive field with a size approximating the parasol receptive field center and then linearly summed; (2) in a spatially nonlinear model, subunit outputs were rectified, weighted by a Gaussian and summed. Unlike the model of *Figure 4*, these models do not incorporate separate excitatory and inhibitory paths, but they do capture the linear vs nonlinear difference in how On and Off parasol RGCs integrate inputs across space.

We used responses to many image patches to define constant–response contours for each of these models in a space spanned by the local luminance and spatial contrast of the sampled patches (patch statistics were measured within the receptive field center, see white circle in *Figure 9B*). These contours identify the set of image properties (i.e., local luminance and spatial contrast) that resulted in similar responses of a given RGC model and hence that cannot be distinguished based on the model response.

As expected, response contours for spatially linear models did not depend on spatial contrast (i.e., the contours are near horizontal; *Figure 9A*, left). The contours for the spatially nonlinear models, however, did depend on spatial contrast because of nonlinear spatial integration (*Figure 9A*, right). Interestingly, the contours for Off parasol cells showed a stronger dependence on spatial contrast than those for On parasol cells. This On/Off difference is not due to an asymmetry in the models, as it was absent for spatial gaussian noise in which the distribution of bright and dark pixels was symmetrical. The difference instead is created by the relative abundance of dark pixels in natural images (*Ratliff et al., 2010*; *Cooper and Norcia, 2015*), and correspondingly could be recapitulated by spatial noise with a skewed pixel distribution (not shown).

Our results suggest that On parasol RGCs exhibit near-linear spatial integration (i.e., top left in *Figure 9A*) while Off parasol RGCs show nonlinear integration (bottom right). In this case, image patches with positive local luminance and high spatial contrast should activate both On and Off parasol cells; such coactivation would be absent if both cells exhibited linear spatial integration. These patches are located at the bright side of edges in an image (red points in *Figure 9B*).

The modeled response contours of On and nonlinear Off parasol cells have different dependencies on changes in mean luminance and contrast for image patches in which the cells are coactive: On parasol model responses are affected only by changes in luminance, while Off parasol model responses depend on both luminance and spatial contrast. This means that the joint responses of On and Off cells should permit discrimination of image patches with small differences in local luminance and/or spatial contrast (*Figure 9—figure supplement 1*); such discrimination is not possible from responses of either cell type considered alone. Furthermore, the linear On/nonlinear Off combination provides better discrimination than a nonlinear On/linear Off combination (*Figure 9—figure supplement 1*) due to the stronger Off responses to spatial structure (*Figure 9A*). Consistent with this modeling, image patches with similar mean luminance within the receptive field center that elicited near-identical measured spike responses from On parasol cells could elicit quite different responses from Off parasol cells (*Figure 9—figure supplement 2*).

This analysis indicates that joint activity of On and Off parasol cells may encode stimulus features that would be ambiguous if both cell types integrated linearly over space. Note that our analysis here focuses entirely on stimuli restricted to the receptive field center. The improved coding of specific stimulus aspects comes with a cost of sensitivity to other stimulus features. For example, with nonlinear

spatial integration, Off parasol cells do not uniquely encode decreases in luminance. We return to this issue in Discussion.

## Discussion

The manner in which RGCs integrate inputs across space has important functional and mechanistic implications. Mechanistically, the linearity of spatial integration is used to classify RGC types and gives insight into the operation of the upstream circuits that provide input to the RGC (*Enroth-Cugell and Robson, 1966*; *Hochstein and Shapley, 1976*; *Demb et al., 1999*; *Demb et al., 2001*). Functionally, nonlinear spatial integration enhances RGC sensitivity to some features of the input at the expense of others (*Schwartz et al., 2012*) and is a central component in many specialized retinal computations (reviewed in *Gollisch and Meister, 2010*). Here, we show that the qualitative nature of spatial integration – that is linear vs nonlinear – is dictated by several nonlinear circuit mechanisms acting in concert. The interaction of these mechanisms to control spatial integration results in a striking stimulus dependence in how On parasol RGCs respond to spatial structure.

### Neural computation and responses to complex stimuli

Artificial stimuli, by design, often activate specific circuit mechanisms while avoiding activation of other mechanisms. But complex stimuli, including natural inputs, recruit multiple circuit mechanisms that act in concert to control circuit outputs. Interactions among these coactive mechanisms pose a fundamental challenge to understanding the operation of neural circuits.

Here, we show that the manner in which On parasol RGCs respond to spatial structure in their inputs is shaped by a combination of rapid adaptive nonlinearities in cone phototransduction (time constant <100 ms) together with rectifying synaptic nonlinearities in the circuits generating excitatory and inhibitory inputs — for example nonlinearities at the bipolar output synapse (*Demb et al., 1999*; *Demb et al., 2001*; *Borghuis et al., 2013*). Thus, for contrast-reversing gratings, cones encountering increases and decreases in light intensity start from different adaptation states and consequently generate responses with different amplitudes and kinetics. These differences in cone inputs to On and Off circuits, coupled with rectification in the circuits generating excitatory and inhibitory input to On parasol cells, favor excitatory input over inhibitory input. Thus, gratings elicit a relatively small ratio of inhibitory input to excitatory input, and a phase shift such that excitatory input increases prior to inhibitory input. These properties contribute to strong responses to the spatial structure of the grating.

Natural inputs lack the periodicity of contrast-reversing gratings; consequently, inhibitory input is both large and temporally in phase with excitatory input, and hence is well poised to decrease sensitivity to fine spatial structure. The opposite is true for Off parasol cells – where the periodic changes in cone adaptation during grating stimuli favor inhibitory input over excitatory input. This provides a clear example, with clear functional consequences (see below), of how several nonlinear mechanisms acting in concert control circuit outputs in unexpected ways.

Changes in rectification at bipolar output synapses can also shape the time course of RGC responses to periodic stimuli (*Ke et al., 2014*). In mouse On sustained RGCs, steady light can produce a sustained depolarization of cone bipolar cells, leading to synaptic depression; transient hyperpolarizing stimuli can relieve this depression and enable subsequent depolarizing stimuli to produce large excitatory input (*Ke et al., 2014*). Parasol RGC responses, however, do not appear to be similarly shaped by synaptic depression in our experiments. The onset of a spatially uniform light step from a constant background elicited large excitatory input and strong spike responses in both On and Off parasol cells (*Figure 4—figure supplement 1*; see also *Turner and Rieke, 2016*), indicating that at the mean luminance levels used here, bipolar cells upstream of parasol RGCs have vesicles available to produce strong synaptic output. Thus, synaptic depression in upstream bipolar cells is not likely to play a major role in the asymmetry seen at the onset of a periodic grating stimulus (e.g., *Figure 1*).

The differences in cone signaling that control the linearity or nonlinearity of On parasol spatial integration are subtle. Yet these relatively small changes in input qualitatively change the computational properties of the circuit. This is one of several examples in which control of computation relies on fine tuning of common circuit elements rather than recruitment of distinct circuits or other more exotic mechanisms (see also *Grimes et al., 2014*). The sensitivity of neural circuits to subtle changes in

input or in operating point will be important in constructing models for neural computation, including artificial neural networks that attempt to match the efficiency of real neural circuits in specific tasks.

## Balanced excitation and inhibition

Cortical neurons typically receive high rates of excitatory and inhibitory inputs which largely cancel – a mode referred to as balanced excitation and inhibition (reviewed by *Isaacson and Scanziani, 2011*). The discovery of *E/I* balance has been an important breakthrough in our understanding of how cortical circuits work, and disruption of *E/I* balance can underlie the aberrant neural signaling characteristic of several brain disorders (reviewed by *Sutula and Dudek, 2007*; *Nelson and Valakh, 2015*).

Signaling in circuits exhibiting *E/I* balance often relies on differences in timing of excitatory vs inhibitory inputs (reviewed by *Isaacson and Scanziani, 2011*). For example, in auditory (*Wehr and Zador, 2003*; *Wu et al., 2008*), somatosensory (*Pouille and Scanziani, 2001*; *Swadlow, 2002*; *Wilent and Contreras, 2005*), and visual cortices (*Liu et al., 2010*), the onset of a sensory stimulus creates a brief time window in which excitatory input exceeds inhibitory input; action potentials are preferentially generated in this window. Several circuit features can contribute to these timing differences and their impact on spike output, including delays in inhibitory signals relative to excitatory signals due to routing through extra synapses (*Pouille and Scanziani, 2001*; *Gabernet et al., 2005*) and the location of excitatory inputs on the dendrites and inhibitory inputs on or close to the soma of a target neuron (*Stokes and Isaacson, 2010*).

Here, we show that the relative timing of excitatory and inhibitory input is highly stimulus dependent. As described above, adaptation in the cones during periodic stimuli like contrast-reversing gratings creates kinetic differences in cones encountering increases and decreases in light intensity. Cones encountering these different polarity stimuli dominate input to the circuits generating excitatory and inhibitory input to On parasol RGCs, and the difference in kinetics provides a time window in which excitatory input substantially exceeds inhibitory input. For nonperiodic stimuli, however, the adaptational state of the cones does not correlate strongly with the change in intensity encountered and cones responding to increases and decreases in light level respond with similar kinetics. Hence, spatial structure in such stimuli creates near-simultaneous changes in excitatory and inhibitory input and little or no spike response.

The circuit features giving rise to this stimulus-dependent control of *E/I* balance are common but also distinctive. The distinct circuit features – notably the sensitivity of the circuits controlling excitatory and inhibitory inputs to different aspects of the circuit input (here increases and decreases in light level) – may help identify other circuits in which a similar control could be at work.

## Asymmetries between On and Off RGCs and neural coding

On and Off RGCs of the same type are classically assumed to respond to similar stimulus features with opposite polarity. Multiple findings show that this picture is too simple, and in fact that corresponding On and Off RGCs can exhibit asymmetries in addition to the polarity of their responses. This includes differences in receptive field size (*Balasubramanian and Sterling, 2009*), contrast sensitivity (*Zaghloul et al., 2003*), rectification (*Turner and Rieke, 2016*), and balance of excitatory and inhibitory inputs (*Murphy and Rieke, 2006*). Some of these asymmetries – for example the smaller receptive fields of Off RGCs – have been suggested to be important in population coding of natural inputs (*Balasubramanian and Sterling, 2009*; *Ratliff et al., 2010*; *Pandarinath et al., 2010*; *Karklin and Simoncelli, 2011*). On/Off asymmetries also differ for different RGC types (*Ravi et al., 2018*), a factor that will need to be considered in efficient coding arguments.

On and Off parasol RGCs both exhibit nonlinear spatial integration in response to contrast-reversing gratings (*Petrusca et al., 2007*; *Crook et al., 2008*). Off parasol RGCs similarly respond nonlinearly to spatial structure in natural images, but unexpectedly On parasol RGCs respond near linearly. The difference in sensitivity to spatial structure in natural images suggests that certain features – specifically image patches that have similar positive luminance and differences in spatial contrast – may be distinguishable based on the joint activity of pairs of On and Off cells even when the responses of the individual cells are ambiguous. Determining whether this increased sensitivity to specific image features, at the expense of sensitivity to others, is advantageous will require considering how the entire RGC population encodes natural inputs. Nonetheless, the symmetry of On and Off parasol RGC responses to contrast-reversing gratings and asymmetry of their responses to natural inputs

emphasizes the complexity of the relationship between the statistics of input stimuli and circuit nonlinearities that control important functional aspects of neural responses.

## Materials and methods
### Tissue preparation and recording
Experimental procedures followed those described previously (*Turner and Rieke, 2016*; *Turner et al., 2018*). In brief, macaque retinas (*Macaca fascicularis*, *Macaca nemistrina*, and *Macaca mulatta*) were obtained through the Tissue Distribution Program of the Regional Primate Research Center at the University of Washington. Pieces of retina attached to the pigment epithelium were stored in ~32–34°C oxygenated (95% $O_2$/5% $CO_2$) Ames medium (Sigma, St Louis, MO) and dark adapted for >1 hr. Pieces of retina were then isolated from the pigment epithelium under infrared illumination and flattened onto poly-L-lysine slides. Once under the microscope, tissue was perfused with oxygenated Ames medium at a rate of ~8 ml/min.

### Electrophysiology
Parasol RGCs were targeted for electrical recordings based on their characteristic soma size and response to a light step. Spike responses were measured in the cell-attached or extracellular configurations using electrodes filled with Ames medium. Synaptic inputs were measured in the whole-cell voltage-clamp configuration using electrodes (2–3 MΩ) filled with a solution containing (in mM): 105 Cs methanesulfonate, 10 Tetraethylammonium chloride (TEA-Cl, a potassium channel blocker), 20 4-(2-hydroxyethyl)-1-piperazineethanesulfonic acid (HEPES buffer), 10 ethylene glycol-bis(β-aminoethyl ether)-N,N,N′,N′-tetraacetic acid (EGTA, a calcium chelator), 2 QX-314, 5 Mg-ATP, 0.5 Tris-GTP (~280 mOsm; pH ~7.3 with CsOH). To isolate excitatory or inhibitory synaptic input, cells were held at the estimated reversal potential for inhibitory or excitatory input of ~−60 and ~+10 mV. These voltages were adjusted for each cell to maximize isolation (see *Cafaro and Rieke, 2013*). Voltages have been corrected for a ~−8.5 mV liquid junction potential.

Retina sensitivity was checked before collecting data based on the contrast sensitivity of On parasol cells. All data collected came from retinas in which a 5% contrast spot modulated at 4 Hz produced at least a 20 spikes/s modulation in the On parasol spike rate. 60–70% of the preparations met this criterion.

### Stimuli
Stimuli were presented and data acquired using custom written stimulation and acquisition software packages Stage (stage-vss.github.io) and Symphony (symphony-das.github.io). Labwide acquisition packages can be found at https://github.com/Rieke-Lab/riekelab-package (*Yu, 2021a* copy archived at swh:1:rev:abdd32f596f57613cb470e4b5328e7d6f678ce5e) and protocols used in this study can be found at https://github.com/Rieke-Lab/turner-package (*Yu, 2021b* copy archived at swh:1:rev:e09ed-136af28ceea83df29d84b6cf661d7361fb3). Details of stimulus presentation followed previous work (*Turner and Rieke, 2016*; *Turner et al., 2018*). Unless otherwise noted, mean light levels produced 4000 isomerizations (R*)/M or L-cone/s, 1000 R*/S-cone/s, and 9000 R*/rod/s; this corresponds to $3x10^{-12}$ sugar-cube inches/rod/s (https://en.wikipedia.org/wiki/Denis_Baylor). Stimuli were restricted to a circular aperture with a diameter equal to twice the standard of a Gaussian fit to responses to the dependence of response on spot size (see *Turner and Rieke, 2016*); this process was repeated for each recorded cell. Spatial contrast of image patches was estimated by dividing the standard deviation of the pixel values in the receptive field center by the mean value. The average spatial contrast of the image patches probed was 0.3, and many patches had spatial contrasts exceeding 0.5.

Images were moved across the retina to simulate eye movements. For *Figure 1*, eye movements were taken directly from those measured in human observers viewing the corresponding image (*Van Der Linde et al., 2009*), resampled to our 60 Hz monitor refresh rate (see *Turner and Rieke, 2016*). For the modeling in *Figure 9*, we simulated eye movements using a random-walk process, with independent steps in *x* and *y*. Steps in the random walk occurred every ms, were discrete, had a size equal to the cone spacing in the model. This diffusive process was interrupted every 300 ms by a saccade-like jump in position. The discrete jumps in position consisted of random displacements in *x* and *y* drawn from a uniform distribution between −100 and +100 cone spacings.

## Clustering of inhibitory responses

Inhibitory synaptic inputs elicited by natural image patches are clustered in *Figure 3*. Input to the clustering algorithm consisted only of the time course of responses to all image patches sampled in recording from a given cell. The first three principal components of these combined responses provided a space for the clustering. Each individual response was projected along these three principal components, and the results were clustered using Matlab's kmeans algorithm.

## Subunit models

Subunit models were used to explore spatial integration. The goal of these models was to explore the impact of specific circuit features on predicted sensitivity to spatial structure rather than explain the responses of a specific cell to specific images.

These models consisted of regular grids of subunits for excitatory and inhibitory pathways. Spatial inputs were filtered through the receptive field of each subunit, the filtered signals were (optionally) passed through a rectifying nonlinearity, subunit outputs were summed to generate the RGC input, and this summed signal was thresholded at 0 to generate a predicted spike output. Subunit nonlinearities were set equal to the average excitatory and inhibitory contrast–response functions (see *Figure 4—figure supplement 1*), and outputs of excitatory subunits were given a threefold larger weight than outputs of inhibitory subunits to reflect the larger driving force associated with excitatory inputs near spike threshold. Subunit receptive field sizes and spacing were set to 1/5th the RGC receptive field center size for excitatory subunits, and inhibitory subunits were 50% larger. These reflect the subunit sizes indicated by responses to contrast-reversing gratings (*Turner and Rieke, 2016*).

Several properties of the inhibitory subunits were altered from this 'base' model for the analyses in *Figure 4*. Inhibitory subunit size and spacing were increased, the positive contrast part of the contrast–response function for inhibitory subunits was set to 0, or the subunit nonlinearity for inhibitory subunits was removed altogether, rendering the subunits linear.

A front-end model of cone phototransduction was added to the model for the analyses in *Figures 6–8*. This model consists of a set of differential equations that capture measured cone responses to a broad range of stimuli (*Angueyra et al., 2022*; *Figure 6—figure supplement 1*). All parameters of the cone model were set by previous measurements. Spatiotemporal inputs were passed through this front-end model, and the predicted cone responses provided input to subunit models constructed as above.

For the analysis of *Figure 9*, parasol models were simplified by removing the inhibitory pathway and making the excitatory pathway linear (for On parasol RGCs) or nonlinear (for Off parasol RGCs). These models captured the essential distinction between linear and nonlinear spatial integration.

## Cone light-adaptation clamp

*Figure 7* uses stimuli designed to minimize nonlinearities in the cone responses. We used two models of the cone responses to identify these stimuli: (1) a full model of the cone responses that captures responses to a broad range of stimuli, including those invoking adaptation; (2) a linear model. The linear model was determined by the response of the full model to a brief, low-contrast flash (i.e., a flash within the linear range of the full model behavior). The stimulus for the full model was a transformed version of the original stimulus, while the original stimulus (untransformed) provided input to the linear model. We then sought a stimulus transformation that minimized the difference between the outputs of the two models. For sinusoidal stimuli, such as the contrast-reversing gratings we used in *Figure 7*, this is particularly simple: the response of the linear model to these stimuli is also sinusoidal, and hence our procedure identifies a stimulus to the full model that creates a sinusoidal output. We refer to this as a 'light-adaptation clamp' because the procedure aims to generate cone responses that lack adaptation.

We identified the appropriate stimulus transformation using a gradient-descent approach. We discretized the stimulus into time bins and then perturbed the stimulus at these discrete times. We retained perturbations that decreased the mean-squared difference between the two models' responses (see *Figure 7A*) using Matlab's fminsearch algorithm. We iterated this process while decreasing the size of the time bins until achieving a stable minimum of the mean-square difference.

## Statistics

Data, where appropriate, are plotted as mean ± SEM. Reported p values are from Wilcoxon rank sum tests.

## Acknowledgements

We thank Mark Cafaro and Shellee Cunnington for excellent technical support. Tissue was provided by the Tissue Distribution Program at the Washington National Primate Research Center (WaNPRC), and we are grateful for assistance from the WaNPRC staff, especially Chris English. Qiang Chen, EJ Chichilnisky, Greg Field, Will Grimes, Mike Manookin, and Greg Schwartz provided invaluable discussions and feedback on an early draft of the manuscript. This work was supported by NIH grants F31-EY026288 (MHT) and EY028542 (FR).

## Additional information

### Competing interests

Fred Rieke: Reviewing editor, *eLife*. The other authors declare that no competing interests exist.

### Funding

| Funder | Grant reference number | Author |
| --- | --- | --- |
| National Institutes of Health | EY028542 | Fred Rieke |
| National Institutes of Health | F31-EY026288 | Maxwell H Turner |

The funders had no role in study design, data collection, and interpretation, or the decision to submit the work for publication.

### Author contributions

Zhou Yu, Conceptualization, Data curation, Formal analysis, Investigation, Writing - review and editing; Maxwell H Turner, Conceptualization, Data curation, Formal analysis, Investigation, Methodology, Writing - original draft, Writing - review and editing; Jacob Baudin, Methodology; Fred Rieke, Conceptualization, Data curation, Formal analysis, Funding acquisition, Investigation, Methodology, Supervision, Writing - original draft, Writing - review and editing

### Author ORCIDs

Maxwell H Turner 
Fred Rieke 

### Ethics

Experiments were performed on primate retina obtained through the Tissue Distribution Program of the University of Washington's Regional Primate Research Center. Recordings were made from retinas from Macaca fascicularis, Macaca nemestrina, and Macaca mulatta of both sexes, aged 2–20 years. All use of primate tissue was in accordance with the University of Washington Institutional Animal Care and Use Committee (protocol 4140-01).

### Decision letter and Author response

Decision letter https://doi.org/10.7554/eLife.70611.sa1
Author response https://doi.org/10.7554/eLife.70611.sa2

## Additional files

### Supplementary files

• Transparent reporting form

## Data availability

Source data for Figures 2, 3, 5, and 7 is provided.

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
