## [Editor Report]

This study provides strong evidence that adaptation dynamics in cone photoreceptors of the primate retina can subtly change the balance of excitatory and inhibitory inputs to On parasol ganglion cells and thereby fundamentally affect how these cells integrate visual information. This provides important mechanistic insight into the previous observation that On parasol cells display nonlinear spatial stimulus integration under standard reversing gratings but linearly integrate signals in the context of natural scenes.

---

## [Decision Letter]

**Decision letter after peer review:**

Thank you for submitting your article "Adaptation in cone photoreceptors contributes to an unexpected insensitivity of On parasol retinal ganglion cells to spatial structure in natural images" for consideration by *eLife*. Your article has been reviewed by 3 peer reviewers, and the evaluation has been overseen by a Reviewing Editor and Tirin Moore as the Senior Editor. The following individuals involved in review of your submission have agreed to reveal their identity: Tom Baden (Reviewer #1); Tim Gollisch (Reviewer #2).

Essential revisions:

There was agreement amongst the reviewers that the study is interesting and novel. However all three revisers felt some restructuring and/or further explanation was needed to better convey the context and interpretation of the findings. I would urge you to read the reviewer comments a make the changes accordingly. You may want to consider some conceptual schematics to increase clarity of a few other arguments.

1. The clarity of the argument comparing gratings vs natural scenes needs to be improved/ Reviewer 3 made a suggestion for restructuring the manuscript to achieve this.

2. It would be great to further expand the description of the cone model.

3. There needs to be further discussion regarding the potential role for BCs in the encoding properties

*Reviewer #1 (Recommendations for the authors):*

– Please add some discussion on possible roles of bipolar cells circuits (or lack thereof). See "public review".

– Please mention the species in the abstract/intro!

– Consider switching the image insets in Figure 2C to the same order as A and B. I can see that as it is now it does not require crossing over the lines linking to the data points, but personally I found this incredibly confusing until I finally realised that they are swapped. I can see that this is noted in the legend, but it would be much easier to read I think if they were simply in the same order throughout.

– Consistent use of colour in figures. Largely this is done well, but there are a few glitches which threw me off. I gather that the "disc" is green, inhibition is red, excitation blue. But then unfortunately these colours are "reused" to also depict other things, such as part of the circuit in 1A, or (most confusingly), the single summary points in 4D. Because the use of colour is so consistent elsewhere this completely threw me off. Consider using symbols?

– Figure 4 took me a long time to understand, and I finally think I know why: There is no scatterplot for the “linear inhibition” condition that would be equivalent to 4B (no FF) and 4C (large subunits). I understand it would just be a predictable scatterplot which is probably why it is not shown, but for the narrative I think it would be really helpful to add that in. Also, perhaps graphically highlight that in 4C the dots fall below unity (a little) which I guess is the main effect that drops the NLI spikes summary in 4D. That also took me a while to spot.

– In some figures I got quite confused about what exactly the datapoints show/summarise in terms of repeated measured (cells/images/actual repeats?). I shall use Figure 2 to illustrate.

– I understand that 2A-C all tested the same set of image patches (how many patches, how many images?). So what then are the error bars showing. Is that the 8 cells? What is the measure? SEM/SD? confidence intervals?

– Then in 2 D, there are more datapoints with errorbars. However, the errorbars are hidden behind the dots to difficult to read. Also, if again each dot is one image patch, why are there now only 11. Is this because only a subset of patches was used? This then in 2E and F reduces to 5. Are these now down to 5 patches? How are the datapoints in A-C, D and E,F related?

– In the introduction section to crossover/feedforward inhibition I think it would be quite valuable to explicitly explain what exactly these two terms mean in the context of on parasols, and what circuits are thought to mediate them, perhaps alongside a suitable schematic.

*Reviewer #2 (Recommendations for the authors):*

My main comment is that more information about the cone adaptation model would be helpful. Is the adaptation implemented as a feedback to the cone activation? With a particular time scale? I'm aware that this information is in a different manuscript, but since the model is so critical here, it's worth pointing out a few details and provide some intuition. In particular, one may wonder whether the adaptation is independent of the asymmetry in the cone response, as the latter seems to be there from the outset and then decreases over cycles of the reversing grating (Figure 6). If so, it may be worth pointing out that the asymmetry even before adaptation may be crucial for providing sufficient crossover inhibition. Relatedly, it would be good to explain the structure of the linear model. Is this just a linear filter? E.g., one may wonder why the "non-adapting" model produces responses that are closer to the steady state under reversing gratings and not to the un-adapted initial response (e.g. Figures 6C, 7D). To better understand this, it would help to better explain how the original model and the non-adapting, linear model differ.

*Reviewer #3 (Recommendations for the authors):*

1. Figure 1 focuses on example cells without quantifying responses in the population. The authors should consider at least reporting the variability of the main response features in the text, if not adding these population data to the figure.

2. The fundamental nonlinearity – for grating stimuli as well as natural images – seems to depend on cones that process a black-to-white transition, as opposed to a gray-to-white transition. Apparently, being in darkness for even ~100 ms is sufficient to alter the cone adaptation state and drive a subsequent excitatory input through the postsynaptic On bipolar cell, that cannot be easily canceled by inhibition at the ganglion cell level.

It is remarkable that this change in adaptation state of the cone is reflected by such a seemingly small effect on the cone membrane current/potential (Figures 6 and 7). It is so small, in fact, that the authors should probably point it out in the figure. This is especially true for the original and transformed traces in Figure 7B – which look nearly identical. In this case, blowing up the trace in an inset might be helpful.

With that said, can the authors rule out downstream mechanisms from contributing to the adaptation effect that is presumed to occur in cones? For example, the On bipolar cells would be maximally hyperpolarized during exposure to black and could be primed to release most strongly at the subsequent step to either gray or white stimuli, after complete removal of any effects of depletion in the presynaptic terminal (see e.g., Ke et al., 2014 for a similar mechanism in rod bipolar cells).

3. In the Abstract, it sounds like there is something truly mysterious about sine-wave gratings. However, as noted above, any stimulus sequence with the conditions of the reversing grating should drive a similar response. As it is, the natural images were not matched to the grating in terms of contrast (apparently), and so the comparison is not quite fair. It would be better to emphasize the main findings in terms of how the sequence of contrast affects linear/nonlinear responses rather than comparing stimulus types that are not matched on important qualities, like contrast.

4. page 6. It was difficult to be convinced that there are only two interesting patterns of inhibition. As it is, there are three clusters – and it is difficult to see how distinct these are – and then only two are focused on, which seems to be a mostly On vs. a mostly Off pathway response during the period the image is presented.

5. Figures 1B, C, it would be better to show the schematic of the grating in the 'null' (sine) phase rather than cosine phase, which was presumably used in all experiments.

6. Figure 3, it could help to point at what is described in the legend – presumably just the response to stimulus onset – not offset, even though that is the most prominent component of the response in some cases (e.g., the inhibition at the offset of the bright spot). It could also help to shade the time of interest (overlapping the stimulus presentation).

7. Figure 3 legend, it was odd not to see some recognition of cluster 2 in the legend.

8. Figure 4, authors should make it clear that there is no dimension of time in the model.

9. Figure 8A, traces show response in pA. What are these? model voltage-clamped cone responses?

10. Figure 9S2, how is spatial contrast defined?

11. page 2, authors should consider adding citation to Demb et al., 2001 J Neurosci paper – which directly investigates the idea that bipolar cells are the nonlinear subunits.

12. page 6, the 'crossover' terminology is retina jargon and could use some additional explanation for a general audience.

13. page 4, it didn't immediately make sense that the NLI could be used for images but not gratings – since a grating is an example of an image.

[Editors’ note: further revisions were suggested prior to acceptance, as described below.]

Thank you for resubmitting your work entitled "Adaptation in cone photoreceptors contributes to an unexpected insensitivity of On parasol retinal ganglion cells to spatial structure in natural images" for further consideration by *eLife*. Your revised article has been evaluated by Tirin Moore (Senior Editor) and a Reviewing Editor.

The manuscript has been improved and no new additional experiments are required. There is one remaining issue that needs to be addressed that has been raised by Reviewer #3. This can be addressed by additional discussion in which you (1) present the two potential models for explaining the asymmetry between On and Off parasol cells in the context of the findings in mouse and (2) discuss whether your findings support the cone vs bipolar cell model

*Reviewer #1 (Recommendations for the authors):*

All previously raised points have been adequately addressed. Congratulations on an excellent manuscript.

*Reviewer #2 (Recommendations for the authors):*

My comments have been fully addressed. The rearrangement of the material regarding grating responses and the additional information about key features of the cone adaptation model, in particular the rapid kinetics, help clarify the story. Altogether, this is an excellent, compelling, and thought-provoking study.

*Reviewer #3 (Recommendations for the authors):*

The authors improved the clarity of the manuscript in revision.

It remains surprising to this reviewer that the asymmetry in On vs. Off parasol cells is not explained partially (or at least to a greater degree) by some difference in On vs. Off bipolar cell synapses. For example, in studies of mouse rod bipolar cells, Ke et al., (2014; Figures 3, 4) showed that ON α cells and AII amacrine cells do not respond to the first cycle of a contrast-reversing spot (transition from gray-> white) but respond robustly to subsequent cycles (transition from black -> white), with the explanation that release is suppressed in the first case because of vesicle depletion at the rod bipolar cell synapses in the presence of a moderate mean luminance (~100-200 R*/rod/s). This seems to be the same behavior observed in On parasol cells, where half the presynaptic bipolar cells see the same stimulus as the spot stimulus in the earlier case.

The alternative is that the results in On parasol cells are explained completely by asymmetric gain control in the cones, and the authors have reasonable evidence for this alternative in Figures 6 and 7. Do the results in Off parasol cells help to disambiguate these models, i.e., the relative importance of nonlinearities in the cones vs. nonlinearities in On cone bipolar synapses? For example, does the linearization of cone responses impact the Off parasol cells in a predictable way? (This is not a request for additional experiments, just a request to consider whether existing data would further support the cone model). One recommendation is to cite the work by Ke et al., (2014) in reference to the alternative model regarding bipolar cells (lines 639/640).

---

## [Author Response]

Essential revisions:There was agreement amongst the reviewers that the study is interesting and novel. However all three revisers felt some restructuring and/or further explanation was needed to better convey the context and interpretation of the findings. I would urge you to read the reviewer comments a make the changes accordingly. You may want to consider some conceptual schematics to increase clarity of a few other arguments.

Thank you for the detailed and constructive reviews. As detailed below, we have made substantial revisions to the text and figures to try to convey our findings more clearly.

1. The clarity of the argument comparing gratings vs natural scenes needs to be improved/ Reviewer 3 made a suggestion for restructuring the manuscript to achieve this.

We have expanded the comparison of gratings and natural scenes in Figure 1 by including full responses (most importantly showing the onset) to contrast-reversing gratings (new Figure 1B). We also expanded the data showing the dependence of grating responses on contrast to include Off parasol cells and moved that panel from Figure 2 to Figure 1. This panel (Figure 1B, right) shows that responses to contrast-reversing gratings are strong at contrasts well within the range of natural images, and we mention that now on page 4, first paragraph. These changes consolidate all of the key empirical properties of gratings in the first figure.

2. It would be great to further expand the description of the cone model.

We have added a schematic showing the phototransduction cascade and the associated differential equations that comprise our model to Figure 6 —figure supplement 1. As part of this change, we also moved the measured cone responses from the supplementary figure into Figure 6, and the model responses from Figure 6 into the supplementary figure. We have expanded the description of the model in the main text (page 12, first paragraph). This includes emphasizing that adaptation in the model, and in real cones, is very rapid.

3. There needs to be further discussion regarding the potential role for BCs in the encoding properties

We have clarified in several places in the paper that rectification at the bipolar synapse is a central component of the circuits that we study. What we find here is that cone adaptation can regulate the impact of the bipolar nonlinearity on RGC responses by shifting the balance between excitatory and inhibitory signaling. This argument was too abstract in the previous version. We now mention this interaction between the cone and bipolar nonlinearities on page 11 (next to last paragraph), page 14 (next to last paragraph), and page 17 (last paragraph).

Reviewer #1 (Recommendations for the authors):– Please add some discussion on possible roles of bipolar cells circuits (or lack thereof).

Thank you – see comment above about the changes we made in this regard.

– Please mention the species in the abstract/intro!

Done

– Consider switching the image insets in Figure 2C to the same order as A and B. I can see that as it is now it does not require crossing over the lines linking to the data points, but personally I found this incredibly confusing until I finally realised that they are swapped. I can see that this is noted in the legend, but it would be much easier to read I think if they were simply in the same order throughout.

Done

– Consistent use of colour in figures. Largely this is done well, but there are a few glitches which threw me off. I gather that the "disc" is green, inhibition is red, excitation blue. But then unfortunately these colours are "reused" to also depict other things, such as part of the circuit in 1A, or (most confusingly), the single summary points in 4D. Because the use of colour is so consistent elsewhere this completely threw me off. Consider using symbols?

Thanks for the suggestion. We have tried to make color consistent throughout. Green is used for the linear equivalent disc, red for inhibitory input, blue for excitatory input. The remaining distinctions (except in Figure 9) are now made using different symbols, line weights or line styles. We agree this should be helpful.

– Figure 4 took me a long time to understand, and I finally think I know why: There is no scatterplot for the “linear inhibition” condition that would be equivalent to 4B (no FF) and 4C (large subunits). I understand it would just be a predictable scatterplot which is probably why it is not shown, but for the narrative I think it would be really helpful to add that in. Also, perhaps graphically highlight that in 4C the dots fall below unity (a little) which I guess is the main effect that drops the NLI spikes summary in 4D. That also took me a while to spot.

Thank you. We have added the linear inhibition panel to Figure 4 (now Figure 4B). We have also made the unity lines (the expectation for linear spatial integration) more visible, and added a gray box to highlight the region where linear and nonlinear spatial integration differ the most. We mention this region in both the main text (page 9, second to last paragraph) and the figure legend.

– In some figures I got quite confused about what exactly the datapoints show/summarise in terms of repeated measured (cells/images/actual repeats?). I shall use Figure 2 to illustrate.– I understand that 2A-C all tested the same set of image patches (how many patches, how many images?). So what then are the error bars showing. Is that the 8 cells? What is the measure? SEM/SD? confidence intervals?

Thanks. We have added additional descriptions in both the text and figure legends about which data comes from example cells and which from populations, as well as how the population data was analyzed.

– Then in 2 D, there are more datapoints with errorbars. However, the errorbars are hidden behind the dots to difficult to read. Also, if again each dot is one image patch, why are there now only 11. Is this because only a subset of patches was used? This then in 2E and F reduces to 5. Are these now down to 5 patches? How are the datapoints in A-C, D and E,F related?

We have clarified in the main text that each point in Figure 2D is data from a single cell, averaged across image patches. And similarly we have clarified how the x-axis bins in Figures 2E and F were generated and why there are 5 points. We have tried to clarify similar issues throughout the paper – as it obviously is important that it is clear what is plotted!

– In the introduction section to crossover/feedforward inhibition I think it would be quite valuable to explicitly explain what exactly these two terms mean in the context of on parasols, and what circuits are thought to mediate them, perhaps alongside a suitable schematic.

Good suggestion. We have added schematics showing the circuitry for each component of inhibitory input to Figure 3A and modified the text accordingly (page 6, last paragraph).

Reviewer #2 (Recommendations for the authors):My main comment is that more information about the cone adaptation model would be helpful. Is the adaptation implemented as a feedback to the cone activation? With a particular time scale? I'm aware that this information is in a different manuscript, but since the model is so critical here, it's worth pointing out a few details and provide some intuition. In particular, one may wonder whether the adaptation is independent of the asymmetry in the cone response, as the latter seems to be there from the outset and then decreases over cycles of the reversing grating (Figure 6). If so, it may be worth pointing out that the asymmetry even before adaptation may be crucial for providing sufficient crossover inhibition. Relatedly, it would be good to explain the structure of the linear model. Is this just a linear filter? E.g., one may wonder why the "non-adapting" model produces responses that are closer to the steady state under reversing gratings and not to the un-adapted initial response (e.g. Figures 6C, 7D). To better understand this, it would help to better explain how the original model and the non-adapting, linear model differ.

Thank you. We have added a schematic of the phototransduction cascade and the differential equations that provide the basis of our model in Figure 6 —figure supplement 1. We have also clarified how the linear model was constructed (page 12, next to last paragraph). We now indicate that adaptation in the cones results from a light-dependent increase in PDE activity; this (importantly) is more rapid than a feedback-based mechanism. We now describe the speed of cone adaptation in the text, and how it shapes the responses to the periodic stimulus. In brief, cone adaptation is sufficiently rapid that signaling gain will follow every cycle of the periodic grating stimuli. Even the amplitude of the response to the first half-cycle of the grating is impacted by adaptation – as the comparison of the linear and adapting cone model responses indicates (now mentioned on page 13, second paragraph). This is part of a new summary of the interaction between adaptation and the balance of excitatory and inhibitory inputs.

Reviewer #3 (Recommendations for the authors):1. Figure 1 focuses on example cells without quantifying responses in the population. The authors should consider at least reporting the variability of the main response features in the text, if not adding these population data to the figure.

We now include population data for F2/F1 responses in On and Off cells, showing standard errors for these estimates. We have also added a description of the variability seen in natural video responses (see page 3). We have decided to not include population data here because these data are described in more detail in (Turner et al., 2016). We mention the quantification in that previous paper in the text now as well.

2. The fundamental nonlinearity – for grating stimuli as well as natural images – seems to depend on cones that process a black-to-white transition, as opposed to a gray-to-white transition. Apparently, being in darkness for even ~100 ms is sufficient to alter the cone adaptation state and drive a subsequent excitatory input through the postsynaptic On bipolar cell, that cannot be easily canceled by inhibition at the ganglion cell level.It is remarkable that this change in adaptation state of the cone is reflected by such a seemingly small effect on the cone membrane current/potential (Figures 6 and 7). It is so small, in fact, that the authors should probably point it out in the figure. This is especially true for the original and transformed traces in Figure 7B – which look nearly identical. In this case, blowing up the trace in an inset might be helpful.

Good points. Indeed, we found the impact of the subtle difference cone signaling interesting. This result emphasizes that these cells operate in a manner in which changes in the balance of excitatory and inhibitory input have a large impact on spike output. We have added a paragraph on page 13 (second paragraph) that emphasizes the speed of cone adaptation (which indeed has a time constant < 100 ms), and how adaptation modulates the cone gain and the I/E balance during the grating. We have also simplified Figure 7B, expanded the time axis, and highlighted with arrows the differences in responses.

With that said, can the authors rule out downstream mechanisms from contributing to the adaptation effect that is presumed to occur in cones? For example, the On bipolar cells would be maximally hyperpolarized during exposure to black and could be primed to release most strongly at the subsequent step to either gray or white stimuli, after complete removal of any effects of depletion in the presynaptic terminal (see e.g., Ke et al., 2014 for a similar mechanism in rod bipolar cells).

We cannot rule out downstream mechanisms entirely, but we do think that our data indicate that adaptation in the cones can account for the majority of the effect. This is based on two lines of argument. The most direct evidence is that removing cone adaptation using our transformed stimuli in Figure 7 removed the time-dependence of responses at grating onset. Specifically, it eliminated the large I/E ratio and small spike count at grating onset, and the phase shift between excitatory and inhibitory input. This manipulation should leave intact downstream mechanisms. The second piece of evidence is that models incorporating adapting cones (but no downstream adaptation) are able to explain the lack of response at the onset of a grating (Figure 6). We have indicated in the text that downstream time-dependent nonlinearities could also contribute, but that our results highlight the importance of cone adaptation (page 14, second to last paragraph).

3. In the Abstract, it sounds like there is something truly mysterious about sine-wave gratings. However, as noted above, any stimulus sequence with the conditions of the reversing grating should drive a similar response. As it is, the natural images were not matched to the grating in terms of contrast (apparently), and so the comparison is not quite fair. It would be better to emphasize the main findings in terms of how the sequence of contrast affects linear/nonlinear responses rather than comparing stimulus types that are not matched on important qualities, like contrast.

Thanks – we have reworded the abstract to emphasize that responses to spatial structure differ across stimuli in an unexpected way rather than to call out sinusoidal gratings (or natural images) specifically.

4. page 6. It was difficult to be convinced that there are only two interesting patterns of inhibition. As it is, there are three clusters – and it is difficult to see how distinct these are – and then only two are focused on, which seems to be a mostly On vs. a mostly Off pathway response during the period the image is presented.

Indeed, we did not mean to imply that the discrete clusters defined by this analysis reflected a true discreteness of the underlying signals. Instead, the clustering was simply meant as a way to organize the data and estimate the contributions of feedforward and crossover inhibitory input to responses to natural image patches. We have revised this section considerably with this point in mind (see in particular third paragraph on page 7).

5. Figures 1B, C, it would be better to show the schematic of the grating in the 'null' (sine) phase rather than cosine phase, which was presumably used in all experiments.

Done (here and in other figures; we were at least consistent if wrong!).

6. Figure 3, it could help to point at what is described in the legend – presumably just the response to stimulus onset – not offset, even though that is the most prominent component of the response in some cases (e.g., the inhibition at the offset of the bright spot). It could also help to shade the time of interest (overlapping the stimulus presentation).7. Figure 3 legend, it was odd not to see some recognition of cluster 2 in the legend.

Added.

8. Figure 4, authors should make it clear that there is no dimension of time in the model.

Done – page 9, end of third paragraph.

9. Figure 8A, traces show response in pA. What are these? model voltage-clamped cone responses?

Yes – those are currents from our cone model. We have clarified that on page 15, first paragraph and in the figure legend.

10. Figure 9S2, how is spatial contrast defined?

Spatial contrast was defined there as the standard deviation of the pixel values in the receptive field center divided by the mean. This is now mentioned in the figure legend and in the Methods (page 20, next to last paragraph).

11. page 2, authors should consider adding citation to Demb et al., 2001 J Neurosci paper – which directly investigates the idea that bipolar cells are the nonlinear subunits.

Done.

12. Page 6, the ‘crossover’ terminology is retina jargon and could use some additional explanation for a general audience.

We added schematics to Figure 3A to illustrate the circuits responsible for crossover and feedforward inhibitory input and to define those terms.

13. Page 4, it didn’t immediately make sense that the NLI could be used for images but not gratings – since a grating is an example of an image.

That text was removed with the rearrangement of the grating data. The problem is that the “disc” response in the NLI is the response to the linear-equivalent disc. For the grating, the linear equivalent disc does not differ from the background (since the mean of the grating is zero). This makes the NLI equal to 1 for all gratings.

[Editors' note: further revisions were suggested prior to acceptance, as described below.]

Reviewer #3 (Recommendations for the authors):The authors improved the clarity of the manuscript in revision.It remains surprising to this reviewer that the asymmetry in On vs. Off parasol cells Is not explained partially (or at least to a greater degree) by some difference in On vs. Off bipolar cell synapses. For example, in studies of mouse rod bipolar cells, Ke et al., (2014; Figures 3, 4) showed that ON α cells and AII amacrine cells do not respond to the first cycle of a contrast-reversing spot (transition from gray-> white) but respond robustly to subsequent cycles (transition from black -> white), with the explanation that release is suppressed in the first case because of vesicle depletion at the rod bipolar cell synapses in the presence of a moderate mean luminance (~100-200 R*/rod/s). This seems to be the same behavior observed in On parasol cells, where half the presynaptic bipolar cells see the same stimulus as the spot stimulus in the earlier case.The alternative is that the results in On parasol cells are explained completely by asymmetric gain control in the cones, and the authors have reasonable evidence for this alternative in Figures 6 and 7. Do the results in Off parasol cells help to disambiguate these models, i.e., the relative importance of nonlinearities in the cones vs. nonlinearities in On cone bipolar synapses? For example, does the linearization of cone responses impact the Off parasol cells in a predictable way? (This is not a request for additional experiments, just a request to consider whether existing data would further support the cone model). One recommendation is to cite the work by Ke et al., (2014) in reference to the alternative model regarding bipolar cells (lines 639/640).

We were also surprised at the prominent role of cone adaptation. We have added a paragraph to the Discussion describing the Ke et al., results and why we think that this mechanism does not contribute strongly to the On vs Off parasol asymmetry. Specifically, both On and Off parasol cells generate robust responses to spatially-uniform transitions from gray to their preferred stimulus. This is true for both steps and sinusoidal stimuli. That indicates that there is not a high level of synaptic depression prior to the onset of a modulated stimulus that shapes excitatory inputs to either On or Off parasol RGCs.

We have tried to be careful to indicate that we think that the cone gain control accounts for the majority of the On/Off asymmetry, but that other mechanisms could certainly contribute. We did not record from Off parasol cells using the stimulus that linearizes cone responses.